# The Surprising Effectiveness of Diffusion Models for Optical Flow and Monocular Depth Estimation

**Saurabh Saxena, Charles Herrmann, Junhwa Hur, Abhishek Kar,**
**Mohammad Norouzi, Deqing Sun, David J. Fleet**[*]

{srbs,irwinherrmann,junhwahur,abhiskar,deqingsun,davidfleet}@google.com

Google DeepMind and Google Research

## Abstract

Denoising diffusion probabilistic models have transformed image generation with their impressive fidelity and diversity. We show that they also excel in estimating optical flow and monocular depth, surprisingly, without task-specific architectures and loss functions that are predominant for these tasks. Compared to the point estimates of conventional regression-based methods, diffusion models also enable Monte Carlo inference, e.g., capturing uncertainty and ambiguity in flow and depth. With self-supervised pre-training, the combined use of synthetic and real data for supervised training, and technical innovations (infilling and step-unrolled denoising diffusion training) to handle noisy-incomplete training data, and a simple form of coarse-to-fine refinement, one can train state-of-the-art diffusion models for depth and optical flow estimation. Extensive experiments focus on quantitative performance against benchmarks, ablations, and the model's ability to capture uncertainty and multimodality, and impute missing values. Our model, DDVM (Denoising Diffusion Vision Model), obtains a state-of-the-art relative depth error of 0.074 on the indoor NYU benchmark and an Fl-all outlier rate of 3.26% on the KITTI optical flow benchmark, about 25% better than the best published method. For an overview see diffusion-vision.github.io

## 1 Introduction

Diffusion models have emerged as powerful generative models for high fidelity image synthesis, capturing rich knowledge about the visual world [21, 48, 55, 62]. However, at first glance, it is unclear whether these models can be as effective on many classical computer vision tasks. For example, consider two dense vision estimation tasks, namely, optical flow, which estimates frame-to-frame correspondences, and monocular depth perception, which makes depth predictions based on a single image. Both tasks are usually treated as regression problems and addressed with specialized architectures and task-specific loss functions, *e.g.*, cost volumes, feature warps, or suitable losses for depth. Without these specialized components or the regression framework, general generative techniques may be ill-equipped and vulnerable to both generalization and performance issues.

In this paper, we show that these concerns, while valid, can be addressed and that, surprisingly, a generic, conventional diffusion model for image to image translation works impressively well on both tasks, often outperforming the state of the art. In addition, diffusion models provide valuable benefits over networks trained with regression; in particular, diffusion allows for approximate inference with multi-modal distributions, capturing uncertainty and ambiguity (*e.g.* see Figure 1).

One key barrier to training useful diffusion models for monocular depth and optical flow inference concerns the amount and quality of available training data. Given the limited availability of labelled

---

[*]DF is also affiliated with the University of Toronto and the Vector Institute.

37th Conference on Neural Information Processing Systems (NeurIPS 2023).

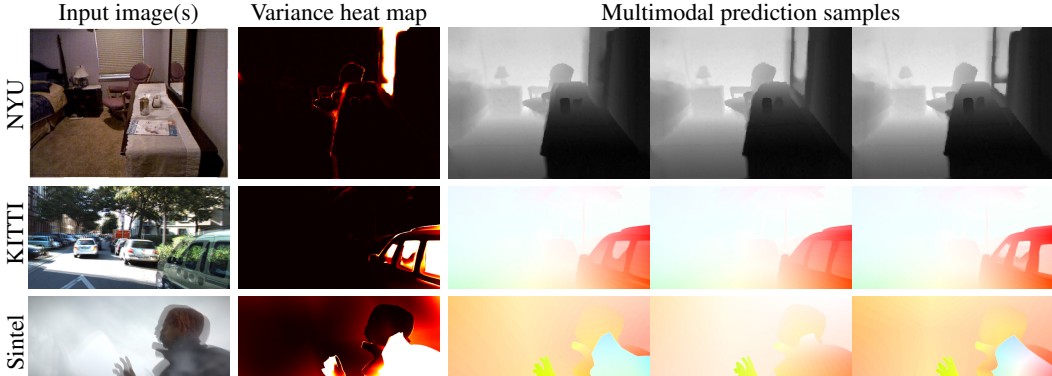

Figure 1: **Examples of multi-modal prediction** on depth (NYU) and optical flow (Sintel and KITTI). Each row shows an input image (or overlayed image pair for optical flow), a variance heat map from 8 samples, and 3 individual samples. Our model captures multi-modality in uncertain/ambiguous cases, such as reflective (*e.g.* mirror on NYU), transparent (*e.g.* vehicle window on KITTI), and translucent (*e.g.* fog on Sintel) regions. High variance also occurs at object boundaries, which are often challenging cases for optical flow, and also partially originate from noisy ground truth measurements for depth. See Fig. 8, 9, 10 and 11 for more examples.

training data, we propose a training pipeline comprising multi-task self-supervised pre-training followed by supervised pre-training using a combination of real and synthetic data. Multi-task self-supervised pre-training leverages the strong performance of diffusion models on tasks like colorization and inpainting [e.g., 54]. We also find that supervised (pre-)training with a combination of real and large-scale synthetic data improves performance significantly.

A further issue concerns the fact that many existing real datasets for depth and optical flow have noisy and incomplete ground truth annotations. This presents a challenge for the conventional training framework and iterative sampling in diffusion models, leading to a problematic distribution shift between training and inference. To mitigate these issues we propose the use of an $L_1$ loss for robustness, infilling missing depth values during training, and the introduction of *step-unrolled denoising diffusion*. These elements of the model are shown through ablations to be important for both depth and flow estimation.

Our contributions are as follows:

1. We formulate optical flow and monocular depth estimation as image to image translation with generative diffusion models, without specialized loss functions and model architectures.
2. We identify and propose solutions to several important issues w.r.t data. For both tasks, to mitigate distribution shift between training and inference with noisy, incomplete data, we propose infilling, step-unrolling, and an $L_1$ loss during training. For flow, to improve generalization, we introduce a new dataset mixture for pre-training, yielding a RAFT [74] baseline that outperforms all published methods in zero-shot performance on the Sintel and KITTI training benchmarks.
3. Our diffusion models is competitive with or surpasses SOTA for both tasks. For monocular depth estimation we achieve a SOTA relative error of 0.074 on the NYU dataset and perform competitively on KITTI. For flow, diffusion surpasses the stronger RAFT baseline by a large margin in pre-training and our fine-tuned model achieves an Fl-all outlier rate of 3.26% on the public KITTI test benchmark, ∼25% lower than the best published method [70].
4. Our diffusion model is also shown to capture flow and depth uncertainty, and the iterative denoising process enables zero-shot, coarse-to-fine refinement, and imputation.

## 2 Related work

Optical flow and depth estimation have been extensively studied. Here we briefly review only the most relevant work, and refer the interested readers to the references cited therein.

**Optical flow.** The predominant approach to optical flow is regression-based, with a focus on specialized network architectures to exploit domain knowledge, *e.g.*, cost volume construction [14, 22, 23, 38, 68, 81, 83, 85], coarse-to-fine estimation [68, 77, 82], occlusion handling [24, 27, 67], or iterative refinement [25, 26, 74], as evidenced by public benchmark datasets [4, 44]. Some

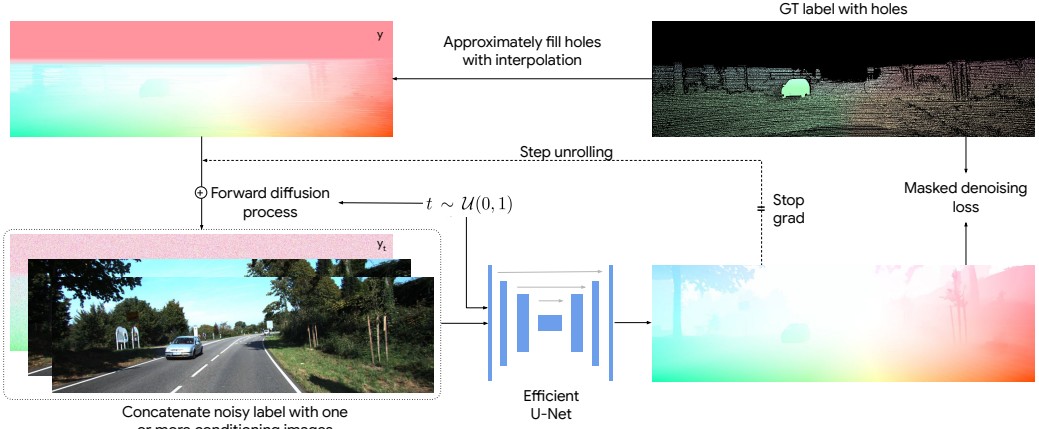

Figure 2: **Training architecture**. Given ground truth flow/depth, we first infill missing values using interpolation. Then, we add noise to the label map and train a neural network to model the conditional distribution of the noise given the RGB image(s), noisy label, and time step. One can optionally unroll the denoising step(s) during training (with stop gradient) to bridge the distribution gap between training and inference for $y_t$.

recent work has also advocated for generic architectures: Perceiver IO [28] introduces a generic transformer-based model that works for any modality, including optical flow and language modeling. Regression-based methods, however, only give a single prediction of the optical flow and do not readily capture uncertainty or ambiguity in the flow. Our work introduces a surprisingly simple, generic architecture for optical flow using a denoising diffusion model.

We find that this generic generative model is surprisingly effective for optical flow, recovering fine details on motion boundaries, while capturing multi-modality of the motion distribution.

**Monocular depth.** Monocular depth estimation has been a long-standing problem in computer vision [58, 59] with recent progress focusing on specialized loss functions and architectures [1, 5, 15, 31] such as the use of multi-scale networks [12, 13], adaptive binning [3, 35] and weighted scale-shift invariant losses [13]. Large-scale in-domain pre-training has also been effective for depth estimation [49, 50, 52], which we find to be the case here as well. We build on this rich literature, but with a simple, generic architecture, leveraging recent advances in generative models.

**Diffusion models.** Diffusion models are latent-variable generative models trained to transform a sample of a Gaussian noise into a sample from a data distribution [21, 62]. They comprise a *forward process* that gradually annihilates data by adding noise, as 'time' $t$ increases from 0 to 1, and a learned *generative process* that reverses the forward process, starting from a sample of random noise at $t = 1$ and incrementally adding structure (attenuating noise) as $t$ decreases to 0. A conditional diffusion model conditions the steps of the reverse process (e.g., on labels, text, or an image).

Central to the model is a denoising network $f_\theta$ that is trained to take a noisy sample $y_t$ at some time-step $t$, along with a conditioning signal $x$, and predict a less noisy sample. Using Gaussian noise in the forward process, one can express the training objective over the sequence of transitions (as $t$ slowly decreases) as a sum of non-linear regression objectives, with the L2 loss (here with the $\epsilon$-parameterization):

$$\mathbb{E}_{(\boldsymbol{x}, \boldsymbol{y})} \, \mathbb{E}_{(t, \boldsymbol{\epsilon})} \left\| f_\theta\left(\boldsymbol{x}, \underbrace{\sqrt{\gamma_t}\, \boldsymbol{y} + \sqrt{1-\gamma_t}\, \boldsymbol{\epsilon}}_{\boldsymbol{y_t}}, \, t\right) - \boldsymbol{\epsilon} \right\|_2^2 \tag{1}$$

where $\boldsymbol{\epsilon} \sim \mathcal{N}(0, I)$, $t \sim \mathcal{U}(0, 1)$, and where $\gamma_t > 0$ is computed with a pre-determined noise schedule. For inference (i.e., sampling), one draws a random noise sample $\boldsymbol{y}_1$, and then iteratively uses $f_\theta$ to estimate the noise, from which one can compute the next latent sample $\boldsymbol{y}_s$, for $s < t$.

**Self-supervised pre-training.** Prior work has shown that self-supervised tasks such as colorization [33, 86] and masked prediction [80] serve as effective pre-training for downstream vision tasks. Our work also confirms the benefit of self-supervised pre-training [54] for diffusion-based image-to-image translation, by establishing a new SOTA on optical flow and monocular depth estimation while also representing multi-modality and supporting zero-shot coarse-to-fine refinement and imputation.

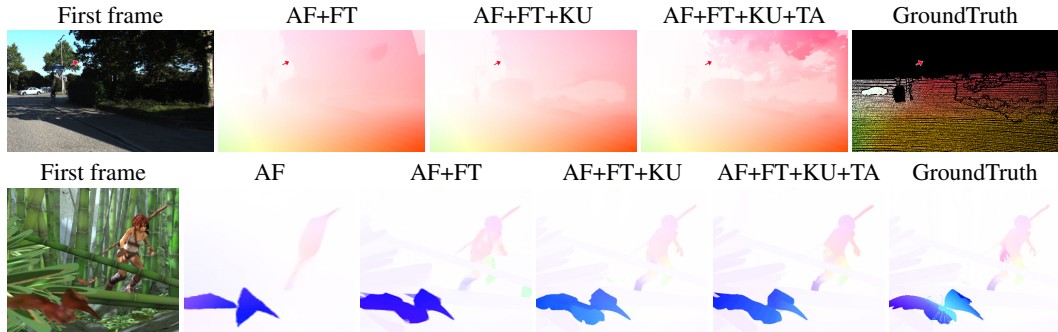

Figure 3: **Effects of adding synthetic datasets in pretraining**. Diffusion models trained only with AutoFlow (AF) tend to provide very coarse flow estimates and can hallucinate shapes. The addition of FlyingThings (FT), Kubric (KU), and TartanAir (TA) remove the AF-induced bias toward polgonal-shaped regions, and significantly improve flow quality on fine detail, e.g. trees, thin structures, and motion boundaries.

## 3 Model Framework

In contrast to the conventional monocular depth and optical flow methods, with rich usage of specialized domain knowledge on their architecture designs, we introduce simple, generic architectures and loss functions. We replace the inductive bias in state-of-the-art architectures and losses with a powerful generative model along with a combination of self-supervised pre-training and supervised training on both real and synthetic data.

The denoising diffusion model (Figure 2) takes a noisy version of the target map (*i.e.*, a depth or flow) as input, along with the conditioning signal (one RGB image for depth and two RGB images for flow). The denoiser effectively provides a noise-free estimate of the target map (*i.e.*, ignoring the specific loss parameterization used). The training loss penalizes residual error in the denoised map, which is quite distinct from typical image reconstruction losses used in optical flow estimation.

### 3.1 Synthetic pre-training data and generalization

Given that we train these models with a generic denoising objective, without task-specific inductive biases in the form of specialized architectures, the choice of training data becomes critical. Below we discuss the datasets used and their contributions in detail. Because training data with annotated ground truth is limited for many dense vision tasks, here we make extensive use of synthetic data in the hope that the geometric properties acquired from synthetic data during training will transfer to different domains, including natural images.

AutoFlow [69] has recently emerged as a powerful synthetic dataset for training flow models. We were surprised to find that training on AutoFlow alone is insufficient, as the diffusion model appears to devote a significant fraction of its representation capacity to represent the shapes of AutoFlow regions, rather than solving for correspondence. As a result, models trained on AutoFlow alone exhibit a strong bias to generate flow fields with polygonal shaped regions, much like those in AutoFlow, often ignoring the shapes of boundaries in the two-frame RGB inputs (*e.g*. see Figure 3).

To mitigate bias induced by AutoFlow in training, we further mix in three synthetic datasets during training, namely, FlyingThings3D [40], Kubric [19] and TartanAir [76]. Given a model pre-trained on AutoFlow, for compute efficiency, we use a greedy mixing strategy where we fix the relative ratio of the previous mixture and tune the proportion of the newly added dataset. We leave further exploration of an optimal mixing strategy to future work. Zero-shot testing of the model on Sintel and KITTI (see Table 1 and Fig. 3) shows substantial performance gains with each additional synthetic dataset.

We find that pre-training is similarly important for depth estimation (see Table 7). We learn separate indoor and outdoor models. For the indoor model we pre-train on a mix of ScanNet [9] and SceneNet RGB-D [41]. The outdoor model is pre-trained on the Waymo Open Dataset [71].

### 3.2 Real data: Challenges with noisy, incomplete ground truth

Ground truth annotations for real-world depth or flow data are often sparse and noisy, due to highly reflective surfaces, light absorbing surfaces [65], dynamic objects [43], *etc*. While regression-based

---

**Algorithm 1** Denoising diffusion train step with infilling and step unrolling

---
1:  $x \leftarrow$ conditioning images, $y \leftarrow$ flow or depth map, $mask \leftarrow$ binary mask of known values
2:  $t \sim U(0, 1), \epsilon \sim N(0, 1)$
3:  $y$ = fill_holes_with_interpolation($y$)
4:  $y_t = \sqrt{\gamma_t} * y + \sqrt{1 - \gamma_t} * \epsilon$
5:  **if** unroll_step **then**
6:      $\epsilon_{pred}$ = stop_gradient($f_\theta(x, y_t, t)$)
7:      $y_{pred} = (y_t - \sqrt{1 - \gamma_t} * \epsilon_{pred})/\sqrt{\gamma_t}$
8:      $y_t = \sqrt{\gamma_t} * y_{pred} + \sqrt{1 - \gamma_t} * \epsilon$
9:      $\epsilon = (y_t - \sqrt{\gamma_t} * y)/\sqrt{1 - \gamma_t}$
10: **end if**
11: $\epsilon_{pred} = f_\theta(x, y_t, t)$
12: loss = reduce_mean($|\epsilon - \epsilon_{pred}|[mask]$)

---

methods can simply compute the loss on pixels with valid ground truth, corruption of the training data is more challenging for diffusion models. Diffusion models perform inference through iterative refinement of the target map $y$ conditioned on RGB image data $x$. It starts with a sample of Gaussian noise $y_1$, and terminates with a sample from the predictive distribution $p(y_0 \,|\, x)$. A refinement step from time $t$ to $s$, with $s < t$, proceeds by sampling from the parameterized distribution $p_\theta(y_s \,|\, y_t, x)$; i.e., each step operates on the output from the previous step. During training, however, the denoising steps are decoupled (see Eqn. 1), where the denoising network operates on a noisy version of the ground truth depth map instead of the output of the previous iteration (reminiscent of teaching forcing in RNN training [79]). Thus there is a distribution shift between marginals over the noisy target maps during training and inference, because the ground truth maps have missing annotations and heavy-tailed sensor noise while the noisy maps obtained from the previous time step at inference time should not. This distribution shift has a very negative impact on model performance. Nevertheless, with the following modifications during training we find that the problems can be mitigated effectively.

**Infilling.** One way to reduce the distribution shift is to impute the missing ground truth. We explored several ways to do this, including simple interpolation schemes, and inference using our model (trained with nearest neighbor interpolation). We find that nearest neighbor interpolation is sufficient to impute missing values in the ground truth maps in the depth and flow field training data.

Despite the imputation of missing ground truth depth and flow values, note that the training loss is only computed and backpropagated from pixels with known (not infilled) ground truth depth. We refer to this as the masked denoising loss (see Figure 2).

**Step-unrolled denoising diffusion training.** A second way to mitigate distribution shift in the $y_t$ marginals in training and inference, is to construct $y_t$ from model outputs rather than ground truth maps. One can do this by slightly modifying the training procedure (see Algorithm 1) to run one forward pass of the model and build $y_t$ by adding noise to the model's output rather than the training map. We do not propagate gradients for this forward pass. This process, called *step-unrolled denoising diffusion*, slows training only marginally (~15% on a TPU v4). This step-unrolling is akin to the predictor-corrector sampler of [63] which uses an extra Langevin step to improve the target marginal distribution of $y_t$. Interestingly, the problem of training / inference distribution shift resembles that of *exposure bias* [51] in autoregressive models, for which the mismatch is caused by *teacher forcing* during training [79]. Several solutions have been proposed for this problem in the literature [2, 32, 84]. Step-unrolled denoising diffusion also closely resembles the approach in [57] for training denoising autoencoders on text.

We only perform step-unrolled denoising diffusion during model fine-tuning. Early in training the denoising predictions are inaccurate, so the latent marginals over the noisy target maps will be closer to the desired *true* marginals than those produced by adding noise to denoiser network outputs. One might consider the use of a curriculum for gradually introducing step-unrolled denoising diffusion in the later stages of supervised pre-training, but this introduces additional hyper-parameters, so we simply invoke step-unrolled denoising diffusion during fine-tuning, and leave an exploration of curricula to future work.

$L_1$ **denoiser loss.** While the $L_2$ loss in Eqn. 1 is ideal for Gaussian noise and noise-free ground truth maps, in practice, real ground truth depth and flow fields are noisy and heavy tailed; *e.g.*, for distant objects, near object boundaries, and near pixels with missing annotations. We hypothesize

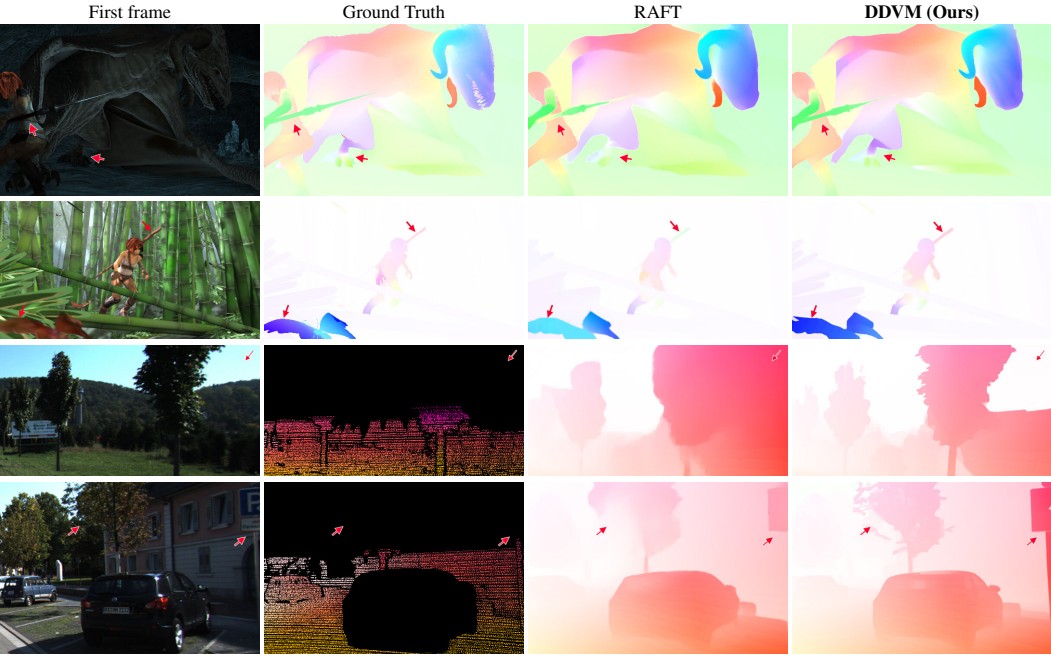

| First frame | Ground Truth | RAFT | DDVM (Ours) |

Figure 4: **Visual results comparing RAFT with our method after pretraining**. Note that our method does much better on fine details and ambiguous regions.

that the robustness afforded by the $L_1$ loss may therefore be useful in training the neural denoising network. (See Tables 11 and 12 in the supplementary material for an ablation of the loss function for monocular depth estimation.)

### 3.3 Coarse-to-fine refinement

Training high resolution diffusion models is often slow and memory intensive but increasing the image resolution of the model has been shown to improve performance on vision tasks [18]. One simple solution, yielding high-resolution output without increasing the training cost, is to perform inference in a coarse-to-fine manner, first estimating flow over the entire field of view at low resolution, and then refining the estimates in a patch-wise manner. For refinement, we first up-sample the low-resolution map to the target resolution using bicubic interpolation. Patches are cropped from the up-scaled map, denoted $z$, along with the corresponding RGB inputs. Then we run diffusion model inference starting at time $t'$ with a noisy map $y_{t'} \sim \mathcal{N}(y_{t'}; \sqrt{\gamma_{t'}} z, (1 - \gamma_{t'})I)$. For simplicity, $t'$ is a fixed hyper-parameter, set based on a validation set. This process is carried out for multiple overlapping patches. Following Perceiver IO [28], the patch estimates are merged using weighted masks with lower weight near the patch boundaries since predictions at boundaries are more prone to errors. (See Section H.5 for more details.)

## 4 Experiments

As our denoiser backbone, we adopt the Efficient UNet architecture [55], pretrained with Palette [54] style self-supervised pretraining, and slightly modified to have the appropriate input and output channels for each task. Since diffusion models expect inputs and generate outputs in the range $[-1., 1.]$, we normalize depths using max depth of 10 meters and 80 meters respectively for the indoor and outdoor models. We normalize the flow using the height and width of the ground truth. Refer to Section H for more details on the architecture, augmentations and other hyper-parameters.

**Optical flow.** We pre-train on the mixture described in Section 3.1 at a resolution of 320×448 and report zero-shot results on the widely used Sintel [4] and KITTI [44] datasets. We further fine-tune this model on the standard mixture consisting of AutoFlow [69], FlyingThings [40], VIPER [53], HD1K [30], Sintel and KITTI at a resolution of 320×768 and report results on the test set from the public benchmark. We use a standard average end-point error (AEPE) metric that calculates L2

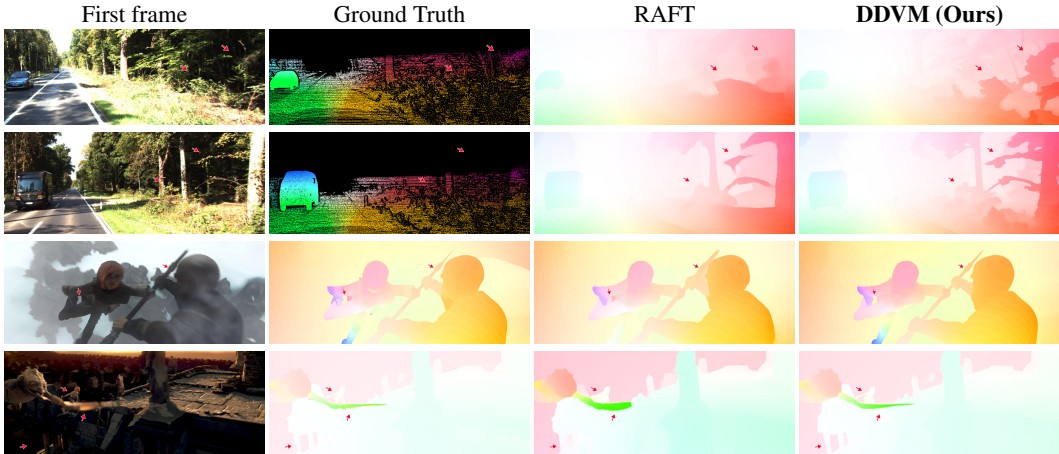

| First frame | Ground Truth | RAFT | **DDVM (Ours)** |

Figure 5: **Visual results comparing RAFT with our method after finetuning**. Ours does much better on fine details and ambiguous regions.

Table 1: **Zero-shot optical flow estimation results on Sintel and KITTI**. We provide a new RAFT baseline using our proposed pre-training mixture and substantially improve the accuracy over the original. Our diffusion model outperforms even this much stronger baseline and achieves state-of-the-art zero-shot results on Sintel.final and KITTI.

| Model | Dataset | Sintel.clean | Sintel.final | KITTI | |
|---|---|---|---|---|---|
| | | AEPE | | AEPE | Fl-all |
| FlowFormer | Chairs→Things | **1.01** | 2.40 | 4.09 | 14.72% |
| RAFT | Chairs→Things | 1.68 | 2.80 | 5.92 | - |
| Perceiver IO | AutoFlow | 1.81 | 2.42 | 4.98 | - |
| RAFT | AutoFlow | 1.74 | 2.41 | 4.18 | 13.41% |
| RAFT (ours) | AF→AF+FT+KU+TA | 1.27 | 2.28 | 2.71 | 9.16% |
| **DDVM (ours)** | AF→AF+FT+KU+TA | 1.24 | **2.00** | **2.19** | **7.58%** |

Table 2: **Optical flow finetuning evaluation** on public benchmark datasets (AEPE↓ for Sintel and Fl-all↓ for KITTI). **Bold** indicates the best and underline the 2nd-best. § uses extra datasets (AutoFlow and VIPER) on top of defaults (FlyingThings, HD1K, KITTI, and Sintel). *uses warm start on Sintel.

| Method | Sintel.clean | Sintel.final | KITTI |
|---|---|---|---|
| SKFlow [72]* | 1.30 | 2.26 | 4.84% |
| CRAFT [66]* | 1.44 | 2.42 | 4.79% |
| FlowFormer [23] | **1.14** | **2.18** | 4.68% |
| RAFT-OCTC [29]* | 1.51 | 2.57 | 4.33% |
| RAFT-it [70]§ | 1.55 | 2.90 | 4.31% |
| **DDVM (ours)**§ | 1.75 | 2.48 | **3.26%** |

distance between ground truth and prediction. On KITTI, we additionally use the outlier rate, Fl-all, which reports the outlier ratio in % among all pixels with valid ground truth, where an estimate is considered as an outlier if its error exceeds 3 pixels and 5% w.r.t. the ground truth.

**Depth.** We separately pre-train indoor and outdoor models on the respective pre-training datasets described in Section 3.1. The indoor depth model is then finetuned and evaluated on the NYU depth v2 dataset [61] and the outdoor model on the KITTI depth dataset [17]. We follow the standard evaluation protocol used in prior work [35]. For both NYU depth v2 and KITTI, we report the absolute relative error (REL), root mean squared error (RMS) and accuracy metrics ($\delta_1 < 1.25$).

## 4.1 Evaluation on benchmark datasets

**Depth.** Table 3 reports the results on NYU depth v2 and KITTI (see Section D for more detailed results and Section B for qualitative comparison with DPT on NYU). We achieve a state-of-the-art absolute relative error of 0.074 on NYU depth v2. On KITTI, our method performs competitively with prior work. We report results with averaging depth maps from one or more samples. Note that most prior works use post processing that averages two samples, one from the input image, and the other based on its reflection about the vertical axis.

**Flow.** Table 1 reports the zero-shot results of our model on Sintel and KITTI Train datasets where ground truth are provided. The model is trained on our newly proposed pre-training mixtures (AutoFlow (AF), FlyingThings (FT), Kubric (KU), and TartanAir (TA)). We report results by averaging 8 samples at a coarse resolution and then refining them to the full resolution as described in Section 3.3. For a fair comparison, we re-train RAFT on this pre-training mixture; this new RAFT model significantly outperforms the original RAFT model. And our diffusion model outperforms the stronger

Table 3: **Performance comparison on the NYU-Depth-v2 and KITTI datasets**. ⊤ indicates method uses unsupervised pretraining, †indicates supervised pretraining and ‡ indicates use of auxilliary supervised depth data. **Best** / second best results are bolded / underlined respectively. ↓: lower is better ↑: higher is better.

| Method | Architecture | NYU-Depth-v2 | | | KITTI | | |
|---|---|---|---|---|---|---|---|
| | | $\delta_1$ ↑ | REL↓ | RMS↓ | $\delta_1$ ↑ | REL↓ | RMS↓ |
| TransDepth [87] | Res-50+ViT-B$^\dagger$ | 0.900 | 0.106 | 0.365 | 0.956 | 0.064 | 2.755 |
| DPT [50] | Res-50+ViT-B$^{\dagger\ddagger}$ | 0.904 | 0.110 | 0.357 | 0.959 | 0.062 | 2.573 |
| BTS [34] | DenseNet-161$^\dagger$ | 0.885 | 0.110 | 0.392 | 0.956 | 0.059 | 2.756 |
| AdaBins [3] | E-B5+Mini-ViT$^\dagger$ | 0.903 | 0.103 | 0.364 | 0.964 | 0.058 | 2.360 |
| BinsFormer [35] | Swin-Large$^\dagger$ | 0.925 | 0.094 | 0.330 | 0.974 | 0.052 | 2.098 |
| PixelFormer [1] | Swin-Large$^\dagger$ | 0.929 | 0.090 | 0.322 | 0.976 | 0.051 | 2.081 |
| MIM [80] | SwinV2-L$^\top$ | 0.949 | 0.083 | 0.287 | **0.977** | **0.050** | **1.966** |
| AiT-P [46] | SwinV2-L$^\top$ | **0.953** | 0.076 | **0.279** | – | – | – |
| **DDVM** samples=1 | Efficient U-Net$^{\top\ddagger}$ | 0.944 | 0.075 | 0.324 | 0.964 | 0.056 | 2.700 |
| **DDVM** samples=2 | Efficient U-Net$^{\top\ddagger}$ | 0.944 | **0.074** | 0.319 | 0.965 | 0.055 | 2.660 |
| **DDVM** samples=4 | Efficient U-Net$^{\top\ddagger}$ | 0.946 | **0.074** | 0.315 | 0.965 | 0.055 | 2.613 |

Table 4: **Ablation on infilling and step-unrolling**. Without either one, performance deteriorates. Without both, optical flow models fail to train on KITTI.

| | NYU val | | KITTI val | | KITTI val | |
|---|---|---|---|---|---|---|
| | REL | RMS | REL | RMS | AEPE | Fl-all |
| Baseline | 0.079 | 0.331 | 0.222 | 3.770 | - | - |
| Step-unroll | 0.076 | **0.324** | 0.085 | 2.844 | 1.84 | 6.16% |
| Infill | 0.077 | 0.338 | 0.057 | 2.744 | 1.53 | 5.24% |
| **Step-unroll & infill** | **0.075** | **0.324** | **0.056** | **2.700** | **1.47** | **4.74%** |

Table 5: **Coarse-to-fine refinement** improves zero-shot optical flow estimation results on Sintel and KITTI, along with the qualitative improvements shown in Figure 6.

| Coarse-to-fine refinement | Sintel.clean | Sintel.final | KITTI | |
|---|---|---|---|---|
| | AEPE | AEPE | AEPE | Fl-all |
| Without | 1.42 | 2.12 | 2.35 | 8.65% |
| **With** | **1.24** | **2.00** | **2.19** | **7.58%** |

RAFT baseline. It achieves the state-of-the-art zero-shot results on both the challenging Sintel Final and KITTI datasets.

Figure 4 provides a qualitative comparison of pre-trained models. Our method demonstrates finer details on both object and motion boundaries. Especially on KITTI, our model recovers fine details remarkably well, *e.g.* on trees and its layered motion between tree and background.

We further finetune our model on the mixture of the following datasets, AutoFlow, FlyingThings, HD1K, KITTI, Sintel, and VIPER. Table 2 reports the comparison to state-of-the-art optical flow methods on public benchmark datasets, Sintel and KITTI. On KITTI, our method outperforms all existing optical flow methods by a substantial margin (even most scene flow methods that use stereo inputs), and sets the new state of the art. On the challenging Sintel final, our method is competitive with other state of the art models. Except for methods using warm-start strategies, our method is only behind FlowFormer which adopts strong domain knowledge on optical flow (*e.g.* cost volume, iterative refinement, or attention layers for larger context) unlike our generic model. Interestingly, we find that our model outperforms FlowFormer on 11/12 Sintel test sequences and our overall worse performance can be attributed to a much higher AEPE on a single (possibly out-of-distribution) test sequence. We discuss this in more detail in Section 5. On KITTI, our diffusion model outperforms FlowFormer by a large margin (30.34%).

## 4.2 Ablation study

**Infilling and step-unrolling.** We study the effect of infilling and step-unrolling in Table 4. For depth, we report results for fine-tuning our pre-trained model on the NYU and KITTI datasets with the same resolution and augmentations as our best results. For flow, we fine-tune on the KITTI train set alone (with nearest neighbor resizing to the target resolution being the only augmentation) at a resolution of 320×448 and report metrics on the KITTI val set [39]. We report results with a single sample and no coarse-to-fine refinement. We find that training on raw sparse data without infilling and step unrolling leads to poor results, especially on KITTI where the ground truth is quite sparse. Step-unrolling helps to stabilize training without requiring any extra data pre-processing. However, we find that most gains come from interpolating missing values in the sparse labels. Infilling and step-unrolling compose well as our best results use both; infilling (being an approximation) does not completely bridge the training-inference distribution shift of the noisy latent.

Table 6: **The addition of optical flow synthetic datasets** substantially improves the zero-shot results on Sintel and KITTI.

| Dataset | Sintel.clean | Sintel.final | KITTI | KITTI Fl-all |
|---|---|---|---|---|
| AF pretraining | 2.04 | 2.55 | 4.47 | 16.59% |
| AF→AF+FT | 1.48 | 2.22 | 3.71 | 14.07% |
| AF→AF+FT+KU | 1.33 | 2.04 | 2.82 | 9.27% |
| AF→AF+FT+KU+TA | **1.24** | **2.00** | **2.19** | **7.58%** |

Table 7: **The addition of synthetic depth data in pre-training** substantially improves fine-tuning performance on NYU.

| Dataset | REL | RMS |
|---|---|---|
| SceneNet RGB-D | 0.089 | 0.362 |
| ScanNet | 0.081 | 0.346 |
| SceneNet RGB-D + ScanNet | **0.075** | **0.324** |

Figure 6: **Visual results with and without coarse-to-fine refinement**. For our pretrained model, refinement helps correct wrong flow and adds details to correct flow.

**Coarse-to-fine refinement.** Figure 6 shows that coarse-to-fine refinement (Section 3.3) substantially improves fine-grained details in estimated optical flow fields. It also improves the metrics for zero-shot optical flow estimation on both KITTI and Sintel, as shown in Table 5.

**Datasets.** When using different mixtures of datasets for pretraining, we find that diffusion models sometimes capture region boundaries and shape at the expense of local textural variation (eg see Figure 3). The model trained solely on AutoFlow tends to provide very coarse flow, and mimics the object shapes found in AutoFlow. The addition of FlyingThings, Kubric, and TartanAir removes this hallucination and significantly improves the fine details in the flow estimates (eg, shadows, trees, thin structure, and motion boundaries) together with a substantial boost in accuracy (*cf*. Table 6). Similarly, we find that mixing SceneNet RGB-D [41], a synthetic dataset, along with ScanNet [9] provides a performance boost for fine-tuning results on NYU depth v2, shown in Table 7.

### 4.3 Interesting properties of diffusion models

**Multimodality.** One strength of diffusion models is their ability to capture complex multimodal distributions. This can be effective in representing uncertainty, especially where there may exist natural ambiguities and thus multiple predictions, *e.g.* in cases of transparent, translucent, or reflective cases. Figure 1 presents multiple samples on the NYU, KITTI, and Sintel datasets, showing that our model captures multimodality and provides plausible samples when ambiguities exist. More details and examples are available in Section A.

**Imputation of missing labels.** A diffusion model trained to model the conditional distribution $p(y|x)$ can be zero-shot leveraged to sample from $p(y|x, y_{partial})$ where $y_{partial}$ is the partially known label. One approach for doing this, known as the *replacement method* for conditional inference [63], is to replace the known portion the latent $y_t$ at each inference step with the noisy latent built by applying the forward process to the known label. We qualitatively study the results of leveraging replacement guidance for depth completion and find it to be surprisingly effective. We illustrate this by building a pipeline for iteratively generating 3D scenes (conditioned on a text prompt) as shown in Figure 7 by leveraging existing models for text-to-image generation and text-conditional image inpainting. While a more thorough evaluation of depth completion and novel view synthesis against existing methods is warranted, we leave that exploration to future work. (See Section C for more details and examples.)

## 5 Limitations

**Latency.** We adopt standard practices from image-generation models, leading to larger models and slower running times than RAFT. However, we are excited by the recent progress on progressive distillation [42, 56] and consistency models [64] to improve inference speed in diffusion models.

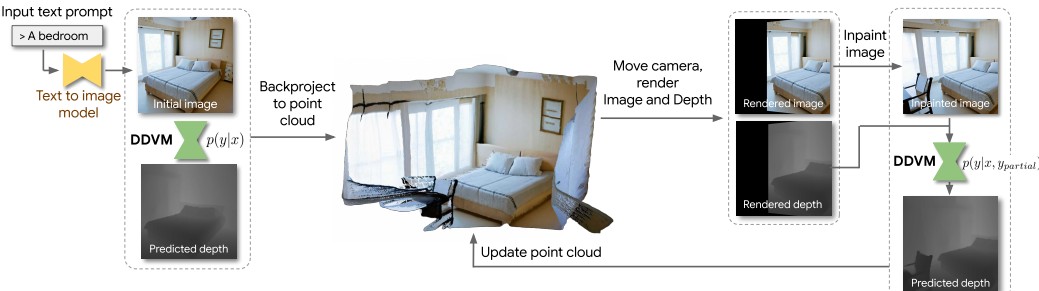

Figure 7: **Application of zero-shot depth completion** with our model by incorporating it into an iterative 3D scene generation pipeline. Starting with a initial image (optionally generated from a text-to-image model), we sample an image-only conditioned depth map using our model. The image-depth pair is added to a point cloud. We then iteratively render images and depth maps (with holes) from this point cloud by moving the camera. We then fill image holes using an existing image inpainter (optionally text conditioned), and then use our model with replacement guidance to impute missing depths (conditioned on the filled RGB image and known depth).

**Sintel fine-tuning.** Under the zero-shot setting, our method achieves state-of-the-art results on both Sintel Final and KITTI. Under the fine-tuning setting, ours is state-of-the-art on KITTI but is behind FlowFormer [23] on Sintel Final. We discuss several possible reasons for why this may be the case.

- We follow the fine-tuning procedure in [70]. While their zero-shot RAFT results are comparable to FlowFormer on Sintel and KITTI, the fine-tuned RAFT-it is significantly better on KITTI but less accurate on Sintel than FlowFormer. It is possible that the fine-tuning procedure (*e.g.* dataset mixture or augmentations) developed in [70] is more suited for KITTI than Sintel.

- Another possible reason is that there is substantial domain gap between the training and test data on Sintel than KITTI. On Sintel test, there is a particular sequence "Ambush 1", where the girl's right arm moves out of the image boundary. Our method has an AEPE close to 30 while FlowFormer has lower than 10. It is likely that the attention on the cost volume mechanism by FlowFormer can better reason about the motion globally and handles this particular sequence well. This particular sequence may account for the major difference in the overall results; among 12 available results on the Sintel website, ours has lower AEPE on 11 sequences but a higher AEPE on the "Ambush 1" sequence, as shown in Table 8. Figure 16 in the appendix further provides visualization.

Table 8: **Average end-point error (AEPE) on 12 Sintel test sequences** available from the public website.

| Sequence | Ours | FlowFormer [23] |
|---|---|---|
| Perturbed Market 3 | **0.787** | 0.869 |
| Perturbed Shaman 1 | **0.219** | 0.252 |
| Ambush 1 | 29.33 | **8.141** |
| Ambush 3 | **2.855** | 2.973 |
| Bamboo 3 | **0.415** | 0.577 |
| Cave 3 | **2.042** | 2.352 |
| Market 1 | **0.719** | 1.174 |
| Market 4 | **5.517** | 8.768 |
| Mountain 2 | **0.176** | 0.518 |
| Temple 1 | **0.452** | 0.612 |
| Tiger | **0.413** | 0.596 |
| Wall | **1.639** | 1.723 |

## 6 Conclusion

We introduced a simple denoising diffusion model for monocular depth and optical flow estimation using an image-to-image translation framework. Our generative approach obtains state-of-the-art results without task-specific architectures or loss functions. In particular, our model achieves an Fl-all score of 3.26% on KITTI, about 25% better than the best published method [70]. Further, our model captures the multi-modality and uncertainty through multiple samples from the posterior. It also allows imputation of missing values, which enables iterative generation of 3D scenes conditioned on a text prompt. Our work suggests that diffusion models could be a simple and generic framework for dense vision tasks, and we hope to see more work in this direction.

**Acknowledgements**

We thank Ting Chen, Daniel Watson, Hugo Larochelle and the rest of Google DeepMind for feedback on this work. Thanks to Klaus Greff and Andrea Tagliasacchi for their help with the Kubric generator, and to Chitwan Saharia for help training the Palette model.

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

# A  Multimodal prediction

We provide more qualitative examples for multimodal prediction. Figures 8 and 9 illustrate multi-modal depth predictions on NYU and KITTI respectively. Multimodality of the posterior distribution exists in regions where there are multiple plausible predictions. For example, this includes reflective and transparent surfaces (mirrors and glass surfaces in rows 1 to 5 of Figure 8 and windows of cars in Figure 9). We further find that the model captures uncertainty in depth estimates in the vicinity of object boundaries, some of which arise due to noise in ground truth measurements in the training data. This can be observed at the boundaries of cars in Figure 9 and around edges of objects in Figure 8 (most clearly visible in the last row).

Figure 10 illustrates different samples on KITTI from the optical flow diffusion model, also capturing multiple modes of the predictive posterior. Multimodality exists on transparent surfaces and near occlusions. As shown in Figure 11, on Sintel, multimodality also exists on occluded or out-of-bounds pixels where multiple predictions are plausible.

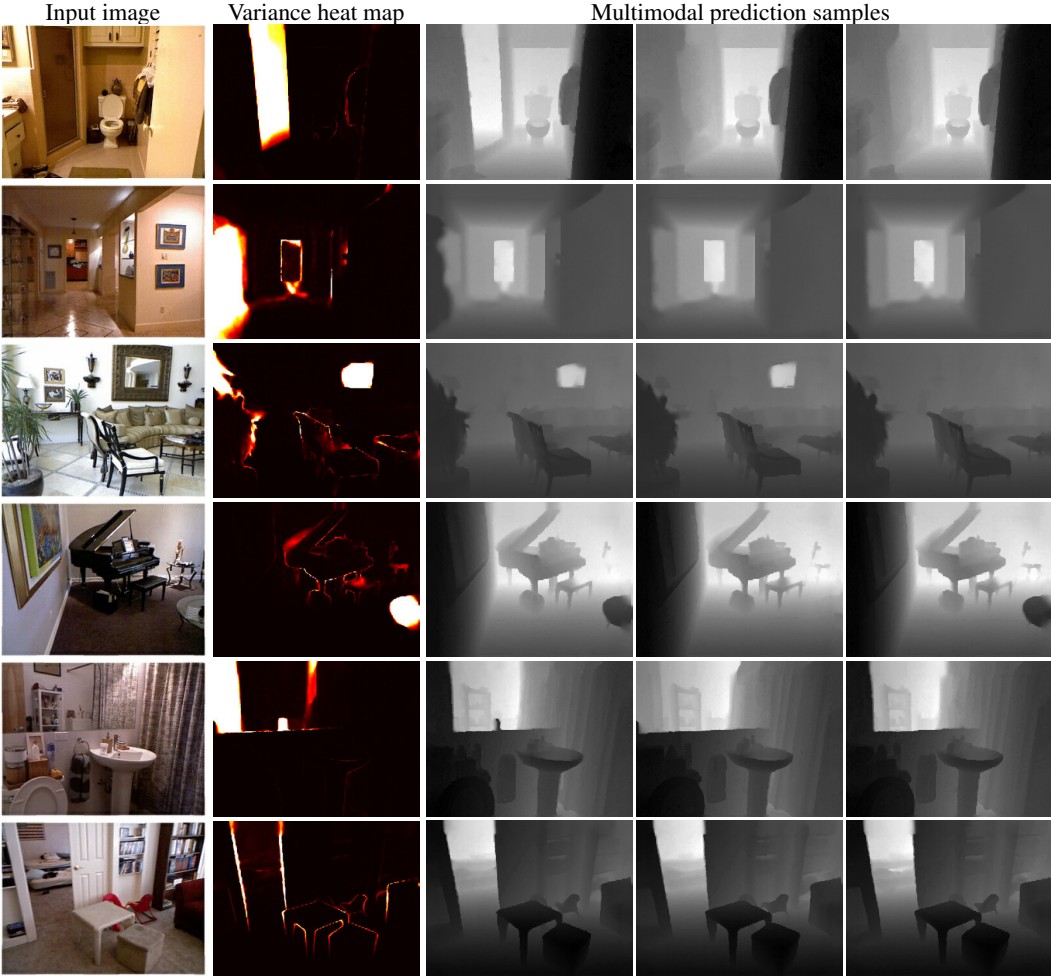

Figure 8: **Qualitative examples of multimodal estimation on the NYU depth dataset**. Our model is able to output multiple plausible depth maps where ambiguity exists. Rows *1 to 5* show transparent or reflective surfaces where two answers exist. In all samples (specially *the last row*) we observe the model's ability to capture uncertainty in depth near object boundaries (see the areas of high variance).

Input image Variance heat map Multimodal prediction samples

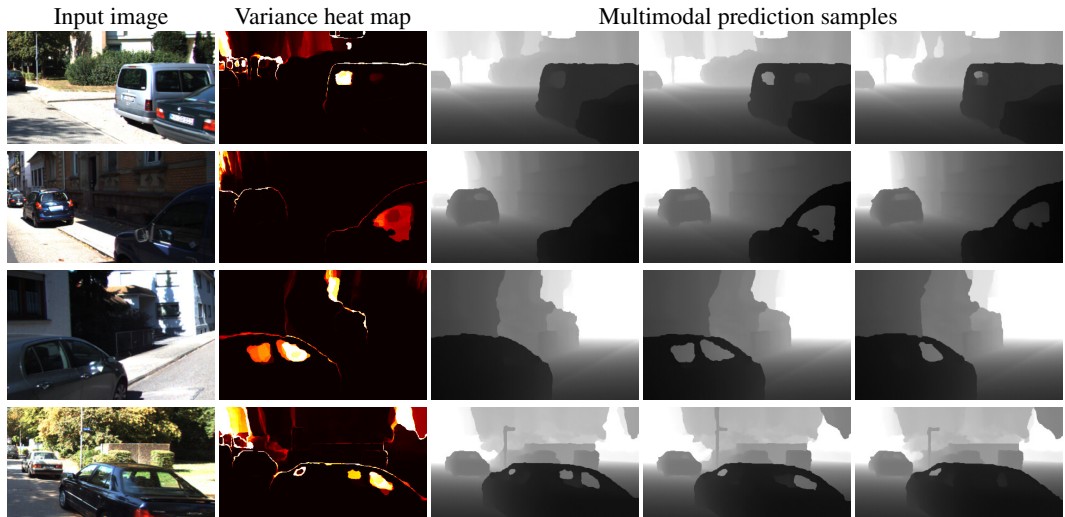

Figure 9: **Qualitative examples of multimodal depth estimation on KITTI**. Our model is able to predict multimodal samples, especially windows of cars and object boundaries. Please refer to the areas with high variance.

Overlayed inputs Variance heat map Multimodal prediction samples

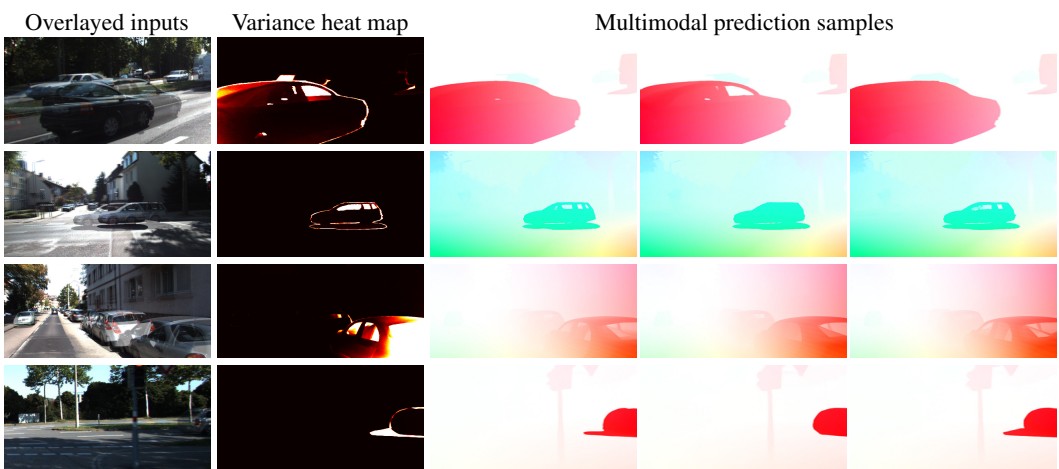

Figure 10: **Qualitative examples of multimodal optical flow estimation on KITTI**. Multimodality exists on transparent surfaces (*e.g*., windows of cars) and shadows where our model estimates layered motion in different samples (*see the last row*).

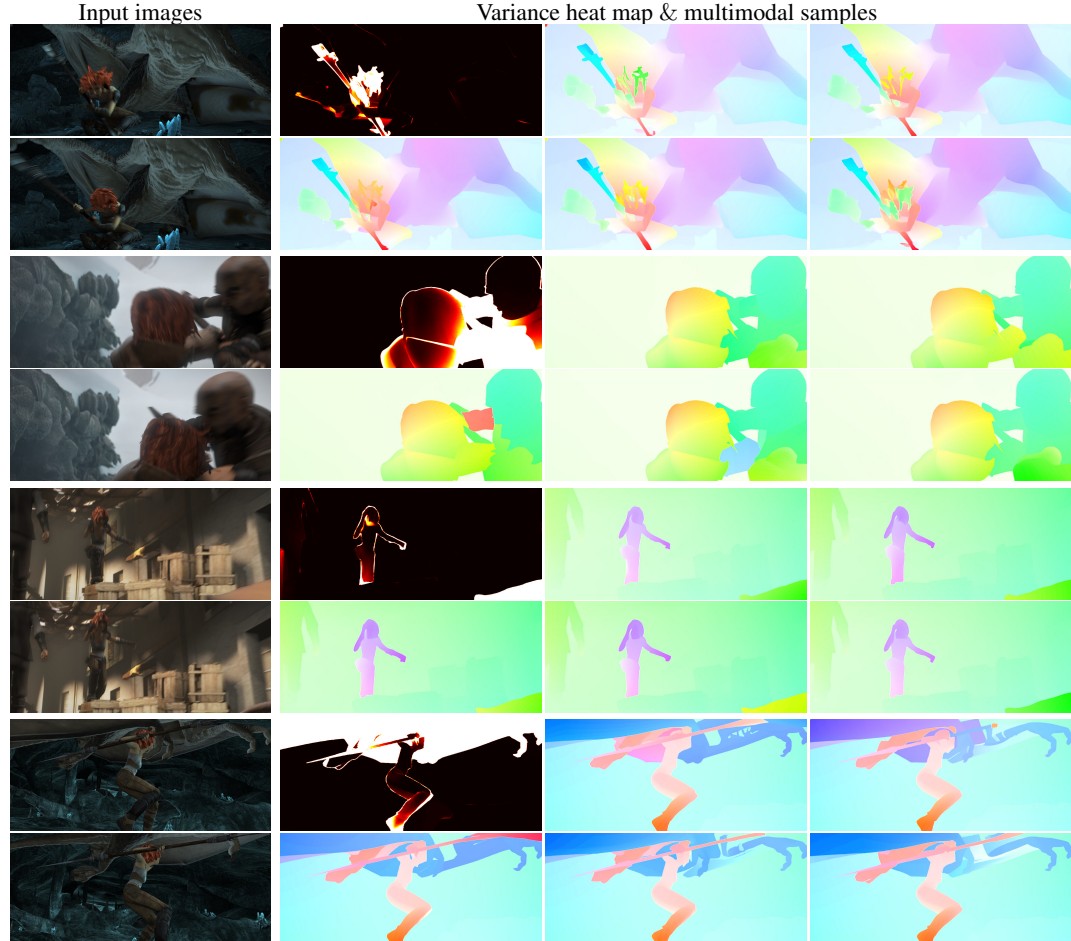

Figure 11: **Qualitative examples of multimodal optical flow estimation on Sintel**. Multimodality also exists on examples with challenging occlusion or out-of-bound cases.

## B   Qualitative comparison of depth estimation with DPT

Figure 12 provides a qualitative comparison of our model with DPT-Hybrid [50] finetuned on the NYU depth v2 [61] dataset. The depth estimates of our diffusion model are more accurate both on coarse-scale scene structure (walls, floors, *etc.*) and on individual objects.

## C   More samples for zero-shot imputation of depth

Figure 13 provides samples generated using our iterative text-to-3D pipeline. We note that such pipelines for iteratively generating 3D scenes have been previously proposed in literature [37, 60, 78]. However, these methods explicitly learn networks to refine the color [37, 60, 78] and the depth map [37, 60]. In contrast, we propose leveraging the text-conditioned image prior from existing large scale text-to-image [55] and text-conditional image completion [75] models, and use our depth estimation model *zero-shot* for depth completion. One caveat with our current approach of using the replacement method for conditional inference [63] for imputing depth, is that it does not enable one to fix errors in the depth predicted in the previous step. One approach to fix artifacts would be by noising-denoising, like that used for coarse-to-fine refinement. We leave further exploration into this to future work.

## D   Complete depth results on NYU and KITTI

Tables 9 and 10 provide detailed results on the val set of NYU depth v2 and KITTI depth datasets. We follow the standard evaluation protocol used in prior work [35]. For both the NYU depth v2

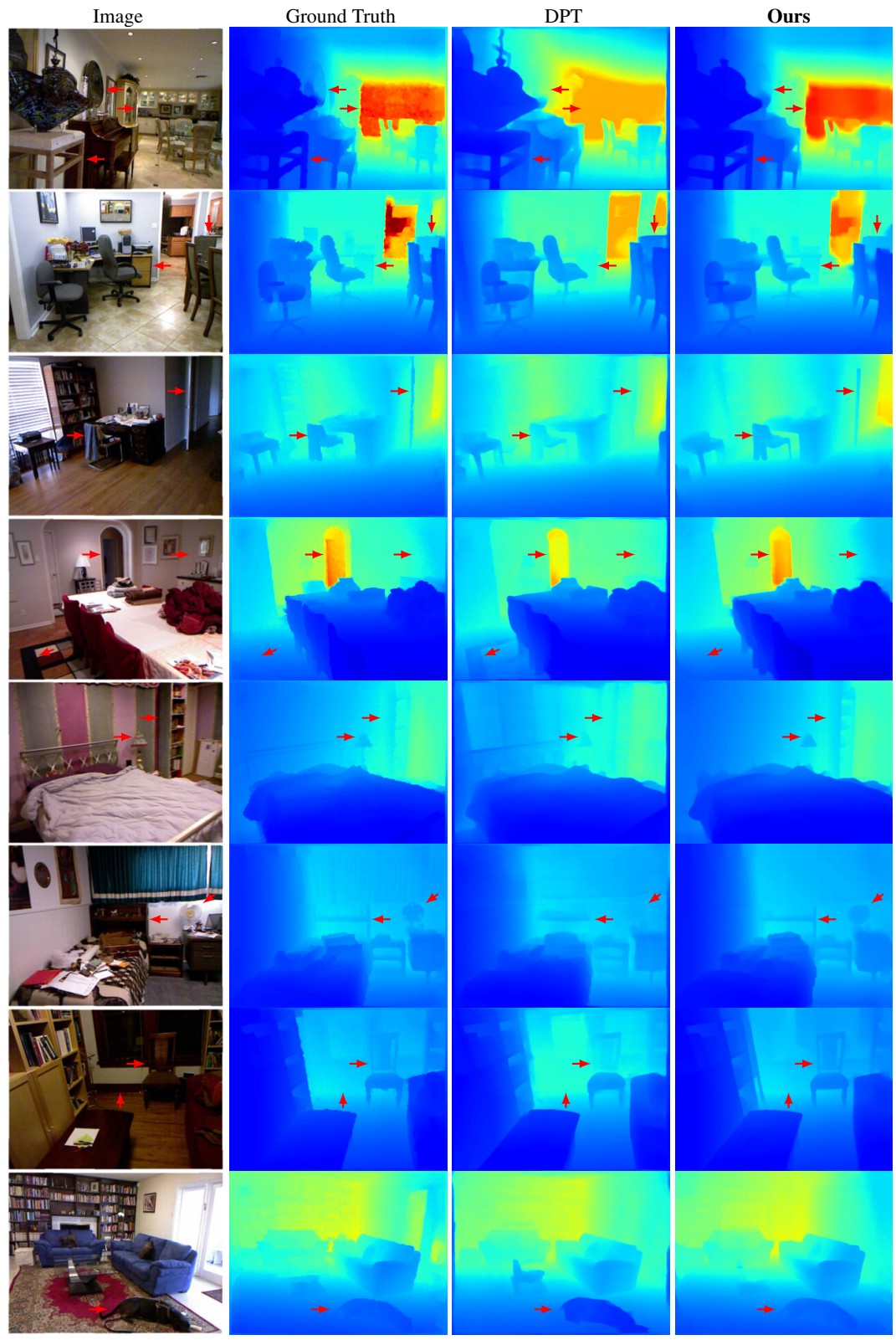

Figure 12: **Qualitative comparison of our model with DPT-Hybrid** [50] (fine-tuned on NYU) on the NYU depth v2 val set. Our method infers better depth for both scene structure (walls, floors, *etc.*) and individual objects. Specific differences are highlighted with red arrows.

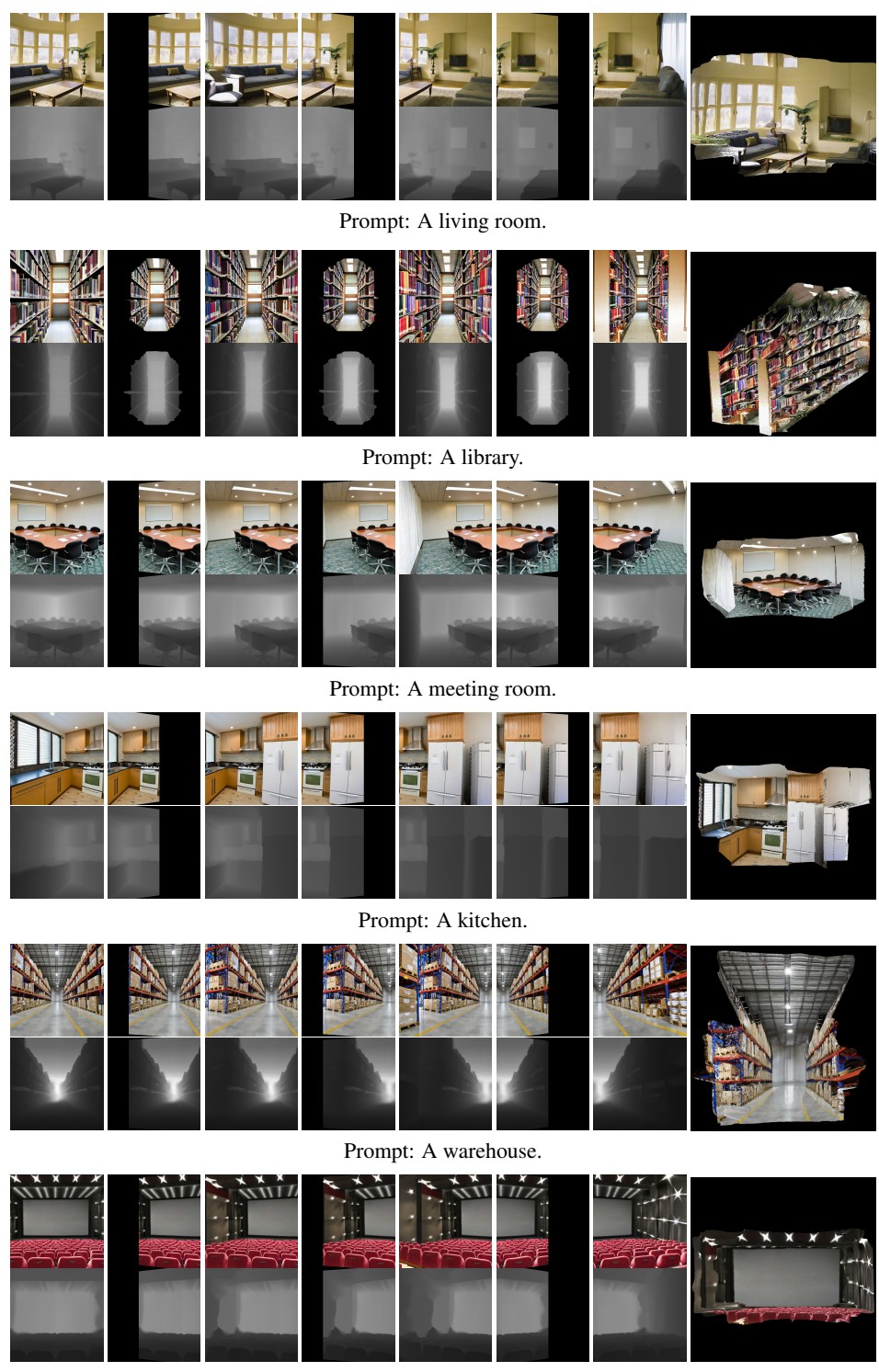

Prompt: A living room.

Prompt: A library.

Prompt: A meeting room.

Prompt: A kitchen.

Prompt: A warehouse.

Prompt: A movie theatre.

Figure 13: **Text-to-3D samples**. Given a text prompt, an image is first generated using Imagen [55] (first row of first column), after which depth is estimated (second row of first column). Subsequently the camera is moved to reveal new parts of the scene which are infilled using an image completion model and our model (which conditions on both the incomplete depth map and the filled image). At each step, newly generated RGBD points are added to a global point cloud which is visualized in the rightmost column.

Table 9: **Comparison of performance on the NYU-Depth-v2 dataset.** ⊤ indicates method uses unsupervised pretraining, † indicates supervised pretraining and ‡ indicates use of auxiliary supervised depth data. **Best /** second best results are bolded / underlined respectively. ↓: lower is better and ↑: higher is better.

| Method | Architecture | $\delta_1 \uparrow$ | $\delta_2 \uparrow$ | $\delta_3 \uparrow$ | REL↓ | RMS↓ | $log_{10} \downarrow$ |
|---|---|---|---|---|---|---|---|
| TransDepth [87] | Res-50+ViT-B$^\dagger$ | 0.900 | 0.983 | 0.996 | 0.106 | 0.365 | 0.045 |
| DPT [50] | Res-50+ViT-B$^{\dagger\ddagger}$ | 0.904 | 0.988 | 0.998 | 0.110 | 0.357 | 0.045 |
| AdaBins [3] | E-B5+Mini-ViT$^\dagger$ | 0.903 | 0.984 | 0.997 | 0.103 | 0.364 | 0.044 |
| BinsFormer [35] | Swin-Large$^\dagger$ | 0.925 | 0.989 | 0.997 | 0.094 | 0.330 | 0.040 |
| PixelFormer [1] | Swin-Large$^\dagger$ | 0.929 | 0.991 | 0.998 | 0.090 | 0.322 | 0.039 |
| MIM [80] | SwinV2-L$^\top$ | 0.949 | **0.994** | **0.999** | 0.083 | 0.287 | 0.035 |
| AiT-P [46] | SwinV2-L$^\top$ | **0.953** | 0.993 | **0.999** | 0.076 | **0.279** | 0.033 |
| **DDVM** samples=1 | Efficient U-Net$^{\top\ddagger}$ | 0.944 | 0.986 | 0.995 | 0.075 | 0.324 | **0.032** |
| samples=2 | Efficient U-Net$^{\top\ddagger}$ | 0.944 | 0.987 | 0.996 | **0.074** | 0.319 | **0.032** |
| samples=4 | Efficient U-Net$^{\top\ddagger}$ | 0.946 | 0.987 | 0.996 | **0.074** | 0.315 | **0.032** |

Table 10: **Comparison of performance on the KITTI dataset.** ⊤ indicates method uses unsupervised pretraining, † indicates supervised pretraining and ‡ indicates use of auxiliary supervised depth data. **Best /** second best results are bolded / underlined respectively. ↓: lower is better and ↑: higher is better. E-B5: EfficientNet-B5 [73].

| Method | Backbone | $\delta_1\uparrow$ | $\delta_2\uparrow$ | $\delta_3\uparrow$ | REL ↓ | Sq-rel ↓ | RMS ↓ | RMS log ↓ |
|---|---|---|---|---|---|---|---|---|
| BTS [34] | DenseNet-161$^\dagger$ | 0.956 | 0.993 | 0.998 | 0.059 | 0.245 | 2.756 | 0.096 |
| TransDepth [87] | ResNet-50+ViT-B$^\dagger$ | 0.956 | 0.994 | 0.999 | 0.064 | 0.252 | 2.755 | 0.098 |
| DPT [50] | ResNet-50+ViT-B$^{\dagger\ddagger}$ | 0.959 | 0.995 | 0.999 | 0.062 | – | 2.573 | 0.092 |
| AdaBins [3] | E-B5+mini-ViT$^\dagger$ | 0.964 | 0.995 | 0.999 | 0.058 | 0.190 | 2.360 | 0.088 |
| BinsFormer [35] | Swin-Large$^\dagger$ | 0.974 | 0.997 | 0.999 | 0.052 | 0.151 | 2.098 | 0.079 |
| PixelFormer [1] | Swin-Large$^\dagger$ | 0.976 | 0.997 | 0.999 | 0.051 | 0.149 | 2.081 | 0.077 |
| MIM [80] | SwinV2-L$^\top$ | **0.977** | **0.998** | **1.000** | **0.050** | **0.139** | **1.966** | **0.075** |
| **DDVM** samples=1 | Efficient U-Net$^{\top\ddagger}$ | 0.964 | 0.994 | 0.998 | 0.056 | 0.339 | 2.700 | 0.091 |
| samples=2 | Efficient U-Net$^{\top\ddagger}$ | 0.965 | 0.994 | 0.998 | 0.055 | 0.325 | 2.660 | 0.090 |
| samples=4 | Efficient U-Net$^{\top\ddagger}$ | 0.965 | 0.994 | 0.998 | 0.055 | 0.292 | 2.613 | 0.089 |

and KITTI datasets we report the absolute relative error (REL), root mean squared error (RMS) and accuracy metrics ($\delta_i < 1.25^i$ for $i \in 1, 2, 3$). For NYU we also report absolute error of log depths ($log_{10}$). For KITTI we additionally report the squared relative error (Sq-rel) and root mean squared error of log depths (RMS log). The predicted depth is up-sampled to the full resolution using bilinear interpolation before evaluation. For the indoor model we evaluate on the cropped region proposed by [13] and for the outdoor model the cropped region proposed by [16] as is standard in prior work.

# E   Ablations

Tables 11 and 12 show that an $L_1$ loss in training the diffusion model performs much better than an $L_2$ loss for monocular depth estimation on NYU and KITTI. Tables 13 and 14 show the effectiveness of Palette-style [54] self-supervised pretraining for monocular depth estimation on NYU and KITTI respectively. All results use a single sample. Because these findings are reasonable and expected to generalize to other dense vision tasks, we do not further ablate them for optical flow estimation for compute efficiency.

Table 11: Ablation for the choice of loss function on the NYU depth v2 dataset.

| | $\delta_1 \uparrow$ | $\delta_2 \uparrow$ | $\delta_3 \uparrow$ | REL↓ | RMS↓ | $log_{10} \downarrow$ |
|---|---|---|---|---|---|---|
| $L_2$ | 0.932 | 0.981 | 0.994 | 0.085 | 0.349 | 0.037 |
| $L_1$ | **0.944** | **0.986** | **0.995** | **0.075** | **0.324** | **0.032** |

Table 12: Ablation for the choice of loss function on the KITTI dataset.

| | $\delta_1\uparrow$ | $\delta_2\uparrow$ | $\delta_3\uparrow$ | REL ↓ | Sq-rel ↓ | RMS ↓ | RMS log ↓ |
|---|---|---|---|---|---|---|---|
| $L_2$ | 0.954 | 0.993 | **0.998** | 0.065 | **0.321** | 2.773 | 0.099 |
| $L_1$ | **0.964** | **0.994** | **0.998** | **0.056** | 0.339 | **2.700** | **0.091** |

Table 13: Ablation for self-supervised pretraining on the NYU depth v2 dataset.

| | $\delta_1 \uparrow$ | $\delta_2 \uparrow$ | $\delta_3 \uparrow$ | REL$\downarrow$ | RMS$\downarrow$ | $log_{10} \downarrow$ |
|---|---|---|---|---|---|---|
| No self-supervised pre-training | 0.936 | 0.980 | 0.992 | 0.081 | 0.352 | 0.035 |
| With self-supervised pre-training | **0.944** | **0.986** | **0.995** | **0.075** | **0.324** | **0.032** |

Table 14: Ablation for self-supervised pretraining on the KITTI depth dataset.

| | $\delta_1 \uparrow$ | $\delta_2 \uparrow$ | $\delta_3 \uparrow$ | REL $\downarrow$ | Sq-rel $\downarrow$ | RMS $\downarrow$ | RMS log $\downarrow$ |
|---|---|---|---|---|---|---|---|
| No self-supervised pre-training | 0.952 | 0.990 | 0.997 | 0.064 | 0.389 | 2.998 | 0.104 |
| With self-supervised pre-training | **0.965** | **0.994** | **0.998** | **0.055** | **0.332** | **2.696** | **0.091** |

## F  Coarse-to-fine refinement for depth

Figure 14 demonstrates performance of coarse-to-fine refinement on the NYU depth v2 dataset. While refinement improves fine-scale details in the estimated depth maps, the qualitative improvements are small and we do not find significant quantitative improvements. Hence the results reported in this work do not use coarse-to-fine refinement for depth estimation. Further work is needed to develop a coarse-to-fine algorithm capable of more robust gains in depth estimation.

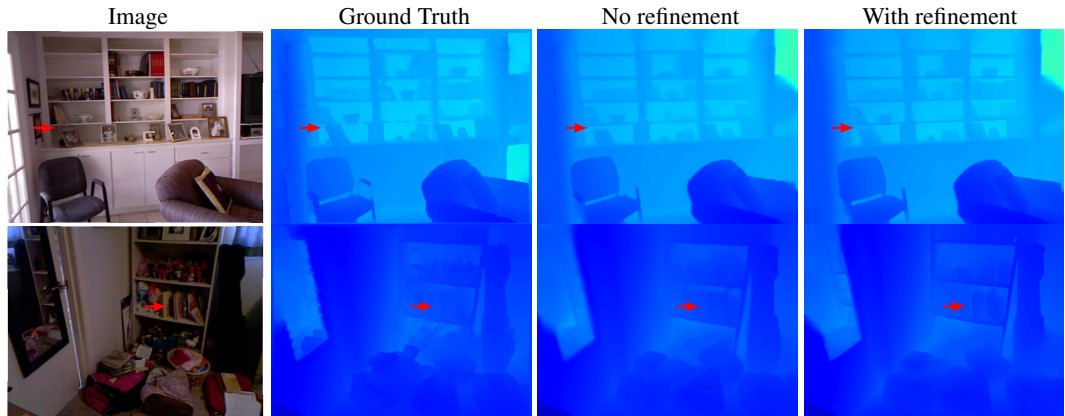

Figure 14: **Samples with coarse-to-fine refinement on the NYU depth v2 dataset.** We find that refinement adds sharpness and detail to the depth estimation but does not provide quantitative improvements.

## G  Coarse-to-fine optical flow refinement for RAFT

For a fair comparison with optical flow estimation, we also apply our coarse-to-fine refinement scheme to RAFT [74], to determine whether our performance gains translate to RAFT as well. We first estimate flow at a low resolution, $320 \times 448$, upsample the low-resolution flow to the original resolution, divide original-resolution input images into $2 \times 5$ overlapping patches of size $320 \times 448$, then estimate flow on the cropped patches using the upsampled flow field as the initial guess for the recurrent refinement (12 steps in total) of RAFT [74]. After estimating flow of each patch, we merge them using weighted masks [28]. Table 15 reports the result. Unlike our diffusion-based method, the coarse-to-fine scheme actually hurts the accuracy of RAFT on Sintel Clean and KITTI and only marginally improves the accuracy on Sintel Final. Further exploration into better approaches for coarse-to-fine refinement for RAFT is warranted. We leave that to future work.

## H  Training and inference details

### H.1  Architecture

**UNet.** The predominant architecture for diffusion models is the U-Net developed for the DDPM model [21], and later improved in several respects [11, 45, 63]. Here we adapt the *Efficient U-Net* architecture that was developed for Imagen [55]. It is more efficient that the U-Nets used in prior work owing to the use of fewer self-attention layers, fewer parameters and less computation at higher

Table 15: **Our coarse-to-fine refinement** scheme marginally improves the performance of RAFT on Sintel Final while hurting performance on Sintel Clean and KITTI. We report the EPE on the Sintel and KITTI datasets.

|  | Sintel Clean | Sintel Final | KITTI |
|---|---|---|---|
| RAFT baseline | 1.27 | 2.28 | 2.71 |
| RAFT with the coarse-to-fine refinement | 1.35 | 2.26 | 2.85 |

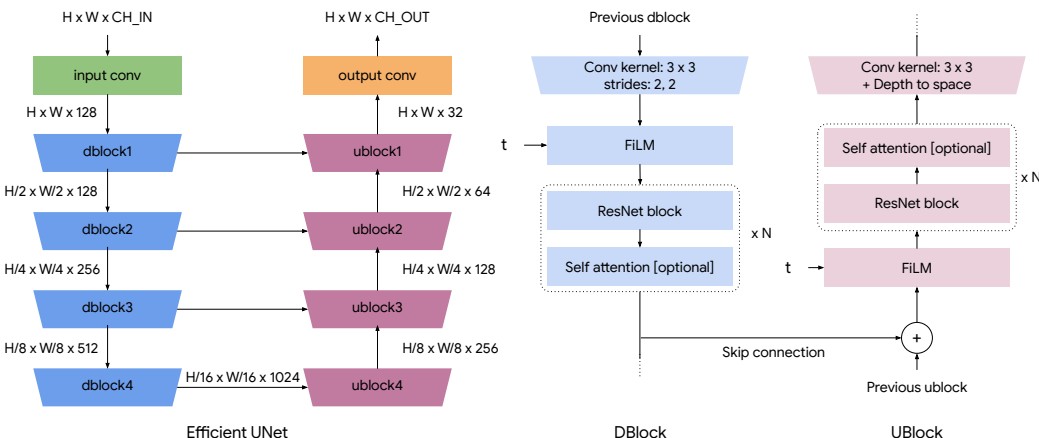

Figure 15: **Overview of the Efficient UNet architecture** proposed in [55]. *CH_IN* and *CH_OUT* refer to the number of input and output channels respectively. *t* refers to the time embedding. *FiLM* refers to the modulation layers proposed in [47]. *N* is the number of ResNet + self-attention blocks.

resolutions, along with other adjustments that make it well suited to training medium resolution diffusion models.

Specifically we adopt the configuration for the $64 \times 64 \rightarrow 256 \times 256$ super-resolution model (see Figure 15 for an overview) with several changes. We drop the text cross-attention layers but preserve the self-attention in the lowest resolution layers *dblock4* and *ublock4* (see Figure 15). For supervised training for the flow model, we find it beneficial to additionally enable self-attention for the last-but-one layers *dblock3* and *ublock3*. The number of input and output channels differ across self-supervised pre-training and supervised pre-training and are also different for flow and depth models. For self-supervised pre-training *CH_IN=6* and *CH_OUT=3* (see Figure 15) since the input consists of a 3-channel source RGB image and a 3-channel noisy target image concatenated along the channel dimension and the output is a RGB image. The supervised depth model has *CH_IN=4* (RGB image + noisy depth) and *CH_OUT=1*. The supervised optical flow model has *CH_IN=8* (2 RGB images + noisy flow along $x$ and $y$) and *CH_OUT=2*. Note that this means we need to reinitialize the input and output convolutional kernels and biases before the supervised pretraining stage. All other weights are re-used.

**Resolution.** Our self-supervised model was trained at a resolution of $256 \times 256$. The indoor depth model is trained at $240 \times 320$. For Waymo we use $256 \times 384$ and for KITTI depth $256 \times 832$. Flow pretraining is done at a resolution of $320 \times 448$, and finetuning at $320 \times 768$.

## H.2 Datasets and augmentation

For unsupervised pre-training, we use the ImageNet-1K [10] and Places365 [88] datasets and train on the self-supervised tasks of colorization, inpainting, uncropping, and JPEG decompression, following [54]. Throughout, we mix datasets at the batch level.

**Flow.** For supervised flow pretraining we use a mix of AutoFlow (native resolution $448 \times 576$), FlyingThings ($540 \times 960$), Kubric ($512 \times 512$) and TartanAir ($480 \times 640$) synthetic datasets. We finetune on the standard mixture consisting of AutoFlow, FlyingThings, Viper ($540 \times 960$), HD1K ($540 \times 1280$), Sintel ($436 \times 1024$), and KITTI ($375 \times 1242$).

We follow the same photometric and geometric augmentation schemes from [70], comprising random affine transformation, flipping, and cropping.

**Depth.** For supervised pre-training of the indoor model we mix the following datasets. *ScanNet* [9] is a dataset of 2.5M examples captured using a Kinect v1-like sensor. It provides depth maps at 480×640 and RGB images at 968×1296. *SceneNet RGB-D* [41] is a synthetic dataset of 5M images generated by rendering ShapeNet [6] objects in scenes from SceneNet [20] at a resolution of 240×320.

For the outdoor model training we use the *Waymo Open Dataset* [71], a large-scale driving dataset consisting of about 200k frames. Each frame provides RGB images from 5 cameras and LiDAR maps. We use the RGB images from the FRONT, FRONT_LEFT and FRONT_RIGHT cameras and the TOP LiDAR only to build about 600k aligned RGB depth maps.

For indoor fine-tuning and evaluation we use *NYU depth v2* [61], a commonly used dataset for evaluating indoor depth prediction models. It provides aligned image and depth maps at 480×640 resolution. We use the official split comprising 50k images for training and 654 for evaluation.

For outdoor fine-tuning and evaluation, we use *KITTI* [17], an outdoor driving dataset which provides RGB images and LiDAR scans at resolutions close to 370×1226. We use the training/test split proposed by [13], comprising 26k training images and 652 test images.

We use random horizontal flip data augmentation which is common in prior work. Where needed, images and dense depth maps are resized using bilinear interpolation to the model's resolution for training and nearest neighbor interpolation is used for sparse maps.

### H.3    Step-unrolling and interpolation of missing depth and flow

As discussed in Section 3.2 of the main paper, infilling and step-unrolling are used to mitigate distribution shift between training and inference with diffusion models. The problem arises due to the missing data in the training depth maps and flow fields.

**Infilling.** For indoor depth maps, we use nearest neighbor interpolation during training (see Section 3.2 in the main paper). For the outdoor depth data we use nearest neighbor interpolation except for sky regions, as they are often large and are much further from the camera than adjacent objects in the image. We use an off-the-shelf sky segmenter [36], and then set all sky pixels to be the maximum modeled depth (here, 80m). For missing optical flow ground truth we employ a simple sequence of 1D nearest neighbor interpolations first along rows, and then along columns.

Table 16: **Ablation of the number of unroll steps for monocular depth estimation**. Performs improves up to four steps of unrolling and plateaus thereafter. The models trained without infilling missing depth benefit more from a larger number of unroll steps, which is to be expected. **Best** and second best results are bolded and underlined respectively.

| | NYU | | | | KITTI | | | |
| --- | --- | --- | --- | --- | --- | --- | --- | --- |
| | No infill | | Infill | | No infill | | Infill | |
| Unroll steps | REL | RMS | REL | RMS | REL | RMS | REL | RMS |
| 0 | 0.079 | 0.331 | 0.077 | 0.338 | 0.222 | 3.770 | 0.057 | 2.744 |
| 1 | 0.076 | 0.324 | 0.075 | 0.324 | 0.085 | 2.844 | 0.056 | 2.700 |
| 2 | 0.076 | 0.317 | 0.075 | **0.315** | 0.068 | 2.799 | 0.054 | 2.629 |
| 3 | **0.075** | **0.316** | **0.074** | 0.317 | 0.061 | 2.789 | 0.054 | 2.591 |
| 4 | **0.075** | **0.316** | **0.074** | **0.315** | 0.059 | **2.739** | **0.053** | **2.568** |

**Step-unrolling.** By default we use a single unroll step in all results where step-unrolling is enabled. In Table 16, we show that using multiple unroll steps can further improve performance on the task of monocular depth estimation.

Finally, while we use infilling and step-unrolling, there are other ways in which one might try to mitigate the problem. One such approach was taken by [46], which faced a similar problem when training a vector-quantizer on depth data. Their approach was to synthetically add more holes following a carefully chosen masking ratio. We prefer our approach since nearest neighbor infilling

is hyper-parameter free and step-unrolled denoising diffusion could be more generally applicable to other tasks with sparse data.

Table 17: **Comparison of step-unrolling and self-conditioning** [8]

| | NYU | | | | KITTI | | | |
| --- | --- | --- | --- | --- | --- | --- | --- | --- |
| | No infill | | Infill | | No infill | | Infill | |
| | REL | RMS | REL | RMS | REL | RMS | REL | RMS |
| Baseline | 0.079 | 0.331 | 0.077 | 0.338 | 0.222 | 3.770 | 0.057 | 2.744 |
| Self conditioning | 0.082 | 0.335 | 0.081 | 0.333 | 0.242 | 3.940 | 0.057 | 2.761 |
| Step unrolling | **0.076** | **0.324** | **0.075** | **0.324** | **0.085** | **2.844** | **0.056** | **2.700** |

We also considered the approach of *self-conditioning* [8] as an alternative to step-unrolling. However, as we show in Table 17, we find that self-conditioning is unable to bridge the train-inference distribution shift of the noisy latent for the task of monocular depth estimation. This is specially apparent in the results for KITTI without infilling where self conditioning leads to no improvement whereas step-unrolling substantially improves performance.

### H.4 Hyper-parameters

**Self-supervised.** The self-supervised model is trained for 2.8M steps with an $L_2$ loss and a mini-batch size of 512. Other hyper-parameters are same as those in the original Palette paper [54].

**Supervised.** The supervised flow and depth models are trained with $L_1$ loss. Usually a constant learning rate of $1 \times 10^{-4}$ with a warm-up over 10k steps is used. However, for depth fine-tuning we find that a lower learning rate of $3 \times 10^{-5}$ achieves slightly better results. All models are trained with a mini-batch size of 64. The indoor depth model is pre-trained for 2M steps and then fine-tuned on NYU for 40k steps. The outdoor depth model is pre-trained for 0.9M steps and fine-tuned on KITTI for 40k steps. For flow, we pretrain for 3.7M steps, followed by finetuning for 50k steps. Other details, like the optimizer and the use of EMA are the same as [54].

### H.5 Inference

**Sampler.** We use the DDPM ancestral sampler [21] with 128 denoising steps for monocular depth models and 64 steps for optical flow models. Increasing the number of denoising steps further did not greatly improve performance.

**Coarse-to-fine refinement.** We use 2×5 overlapping patches ({top, bottom} × {left, center-left, center, center-right, right}) for coarse-to-fine refinement. For Sintel we use $t' = 32/64$ and for KITTI $t' = 8/64$.

## I  Limitations

Table 18: **Inference speed comparison** of our method with DPT [50] on the indoor depth model finetuned on NYU. Diffusion model inference is bottlenecked by the large number of denoising steps. We show that some efficiency gains can be achieved by simply reducing the number of denoising steps. Our model with 24 denoising steps is comparable in performance to DPT while being ∼5x slower (modulo differences in hardware). * we use the step-time reported in the DPT paper at a resolution of $384 \times 384$, however, the DPT performance metrics on NYU are with a model trained at a resolution of $480 \times 640$, for which the step time will be higher.

| Method | Architecture | Resolution | Total Time [ms] | Inference steps | REL ↓ | RMS ↓ |
| --- | --- | --- | --- | --- | --- | --- |
| DPT-Hybrid | Nvidia RTX 2080 | $384 \times 384$* | 38* | - | 0.110 | 0.357 |
| **DDVM** | TPU v4 | $240 \times 320$ | 204 | 24 | 0.104 | 0.378 |
| | | | 272 | 32 | 0.086 | 0.342 |
| | | | 544 | 64 | 0.077 | 0.324 |
| | | | 1089 | 128 | 0.075 | 0.324 |

**Efficiency.** Inference speed with diffusion models is a well-known issue, as multiple denoising steps are used to transform noise to a target signal. This can be prohibitive for vision tasks where near

real-time latency is often desired. Table 18 compares the inference speed of our diffusion model for depth against DPT [50]. Despite having an efficient denoiser backbone (∼8.5 ms per denoising step on a TPU v4), the diffusion model is considerably slower than DPT in total wall time. The most obvious way to reduce inference latency is to reduce the number of denoising steps. This can be done with only moderate reduction in performance. As shown in Table 18, we perform comparably with DPT with as few as 24 denoising steps. However, a more thorough study into optimizing the inference speed of these models while preserving the generation quality is warranted. With the use of progressive distillation [42, 56] it is likely possible to reduce latency even further, as this approach has been shown to successfully distill generative image models with over 1000 denoising steps into those with just 2-4 steps.

**Fine-tuning on Sintel.** In Section 5 we discuss possible reasons for why our model's superior zero-shot performance compared to FlowFormer [23] does not transfer to fine-tuning on Sintel. Figure 16 provides qualitative examples to further support the claims.

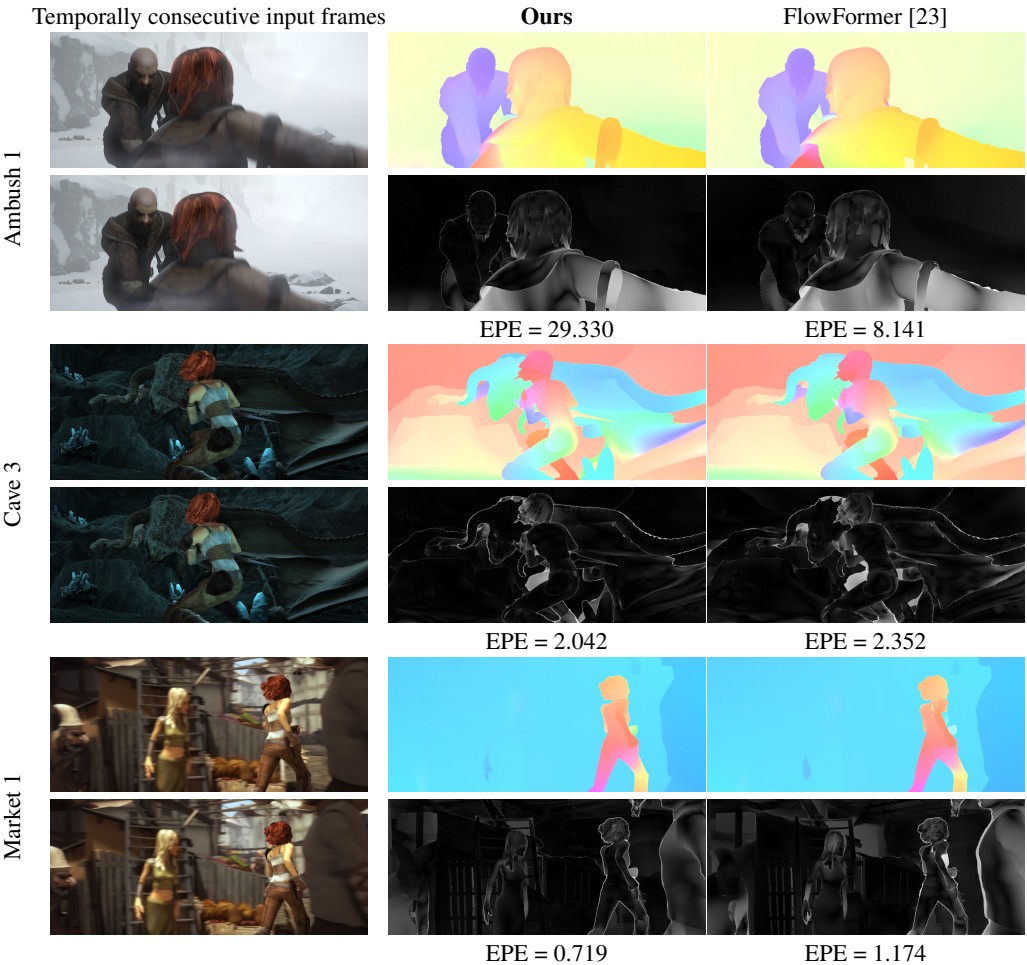

Figure 16: **Visual results on Sintel test**. We compare with Flowformer [23] and provide flow visualization and an error map on each scene. On Ambush 1, FlowFormer can better predict the motion of the girl's right arm that moves out of the image boundary, likely due to the global reasoning capability of attention. On Cave 3 and Market 1, our method provides much finer details on motion boundaries with lower end-point error (EPE).

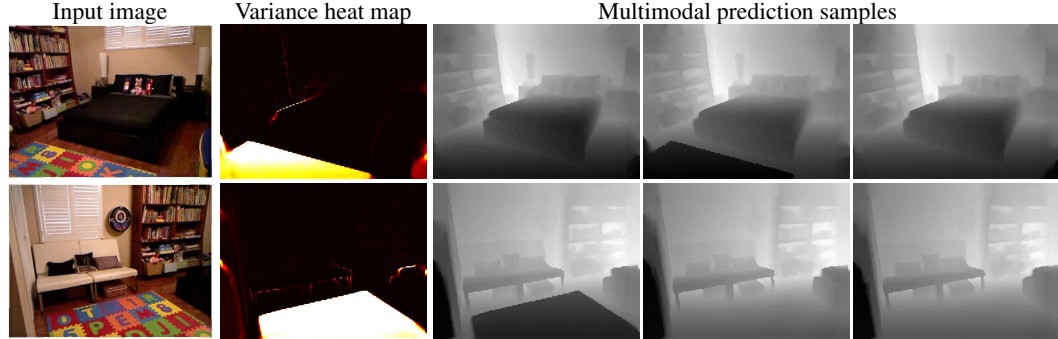

Figure 17: **Qualitative examples of multimodal estimation on the NYU depth dataset** showing examples where our model's *uncertainty* gets captured in the multimodal posterior. In the examples above, the model confuses the play mat (farther away from the viewpoint) for a table (closer to the viewpoint).

**Uncertainty in depth estimation.** We observe certain cases where the model is uncertain about the depth estimates. Interestingly, this uncertainty appears to be well captured in the predictive posterior, as illustrated in Figure 17.

