# OpenReview forum: "The Surprising Effectiveness of Diffusion Models for Optical Flow and Monocular Depth Estimation"
_NeurIPS.cc/2023/Conference — NeurIPS 2023 oral_

### Official Review · Reviewer_LqWs · 2023-06-29

**Soundness:** 4 excellent
**Presentation:** 4 excellent
**Contribution:** 3 good
**Rating:** 7
**Confidence:** 4

**Summary:**

This paper approaches the task of predicting optical flow from a pair of images and depth from a single image. It proposes to do so using diffusion models, providing a training pipeline which includes contributions to deal with noisy training data. The proposed method is competitive with SOTA in depth prediction on NYU and KITTI, is similar or better to SOTA in optical flow prediction on Sintel and KITTI zero-shot, and is SOTA in optical flow after finetuning on KITTI. Additional experiments show the proposed pretraining pipeline meaningfully improves RAFT and the proposed model, that contributions dealing with noisy training data are helpful, and that the proposed method can predict multimodal outputs in cases of uncertainty.

Edit: Thanks to the authors for the rebuttal.

After reading the rebuttal and other reviews I will keep my rating at 7 - accept. I believe this paper should be accepted because it (1) introduces a simple but interesting method that performs SOTA on competitive benchmarks and (2) is very polished and experiments clearly defend all main contributions of the method.

**Strengths:**

The paper provides a simple method that is similar-to or better than SOTA on competitive optical flow and depth tasks
- A new data pipeline is introduced which significantly boosts performance of prior SOTA e.g. RAFT, while improving further with the proposed method
- The method is built off a standard image-to-image diffusion method Palette, adds minimal detail needed to handle flow and depth data (e.g. L1 loss, infilling + unrolling, coarse-to-fine), and is competitive across a wide variety of experiments
- Real data is tricky to handle for diffusion models as it can contain incomplete flow and depth maps. The proposed method has an interesting idea of using its own predictions late in training. Combined with infilling, this is helpful.

The paper is very polished and experiments clearly defend all main contributions of the method.
- Writing is very clear, figures and tables are attractive and helpful
- Choice of training data is very important (Figure 3, Table 6, 7); it even improves RAFT meaningfully (Table 1)
- The method can also produce Multimodal samples faced with ambiguity e.g. transparent/translucent/reflective (Figure 1, 4, 5)
- Figure 6 and Table 5 shows the importance of coarse-to-fine refinement, which enables the method to produce better detail than RAFT (Figure 4, 5)
- Infilling + step-unrolling yields massive improvement on KITTI (Table 4)
- L1 loss (Supp Table 4), pretraining for depth (Supp Table 5, 6)
- Zero-shot depth completion is a cool application of the method

**Weaknesses:**

Finetuning performance on real data yields weaker performance on Sintel vs. other methods (minor weakness, but addressing could further strengthen paper!)
- On Sintel (Table 2), most other methods use warm start, i.e. initializing flow prediction from previous frame. Is it possible to do this with diffusion, i.e. starting from a previous prediction, perhaps denoising fewer steps? If not, this is an important weakness of diffusion models in this setting
- Given most other method warm-start, it is harder to analyze the ability of the proposed method to finetune on real data. Perhaps it does not do as well relative to FlowFormer because (1) it does not handle real data as well as regression methods, even after contributions in this area, or (2) it does not overfit as well to specific datasets given others tend to have task-specific architecture. A comparison to more methods without warm-start, or by using the proposed method with warm-start, it would be very interesting to see this analyzed.

The main novel contribution of this paper beyond data pipeline is step-unrolled denoising diffusion training, which yields modest improvement over infilling alone (minor weakness)
- L1 loss is a design decision, infilling holes using bilinear interpolation is not a substantial contribution in my eyes. So the remaining technical contribution is unrolling.
- I understand the contributions are deliberately simple, which is a positive given performance gain. However, it is still important to analyze the reasons for the method’s success. In this case, the most novel component, “unroll”, improves optical depth REL from 0.077 to 0.075, RMS from 0.338 to 0.324, KITTI REL from 0.057 to 0.056, RMS from 2.744 to 2.700, AEPE from 1.53 to 1.47, and F1-all from 5.24% 4.74%. These contributions are nontrivial, but not substantial on their own.

**Questions:**

- L179 is a bit confusing and could be rewritten “Training high resolution diffusion models is often slow and memory intensive but model performance has been shown to improve with resolution”
- Figure 4 (top) is dark and indistinct, making it a hard to see. Are there failure cases for RAFT on more easily readable examples?

**Limitations:**

Yes

---

> ### Author Rebuttal · Authors · 2023-08-10
>
> Thank you for your review, and for your thoughtful comments and questions. They will certainly help improve the revised paper. Please see our response below.
>
> > Finetuning performance on real data yields weaker performance on Sintel vs. other methods (minor weakness, but addressing could further strengthen paper!)
>
> In Table 9 of the supplementary material, we show that we outperform FlowFormer on all Sintel test sequences but one, ambush_1. This sequence has significant ambiguity due to an all-white background, large camera and object motions, and the existence of numerous objects that occlude one another and move out of frame. This forces the model to depend heavily on inductive biases and learned motion priors, which could be challenging for our method. It is possible that FlowFormer’s inductive bias (with the cost volume etc) make it better suited to reason about this sequence’s large out-of-frame motions. In general, we agree with the reviewer that our model’s worse performance on Sintel compared to models such as FlowFormer, which we outperform by a large margin on KITTI, is surprising and deserves further exploration.
>
> > Is it possible to do warm start with diffusion
>
> There are a couple of ways of doing warm start with diffusion. One would be to use the warped flow as the initial estimate and perform partial denoising as the reviewer suggests. Another would be to guide the denoising process (similar to classifier guidance [19]) using a loss that ensures cross frame consistency. These are interesting directions, but we have not explored them in detail to date.
>
> > (1) it does not handle real data as well as regression methods, even after contributions in this area, or (2) it does not overfit as well to specific datasets given others tend to have task-specific architecture.
>
> As the reviewer has noted, our model achieves SOTA performance on KITTI (a real dataset) by a large margin (Table 2). Relative to FlowFormer, our model’s weaker test performance on Sintel is somewhat surprising. This may be attributable to the differences in the data used for pre-training the Flowformer model vs our diffusion model, or to the presence of a task-specific bias in the FlowFormer architecture. We agree with the reviewer that this is worthy of future exploration.
>
> > Minor weaknesses: 1) L1 loss is a design decision, infilling holes using bilinear interpolation is not a substantial contribution in my eyes. So the remaining technical contribution is unrolling. 2) These contributions are nontrivial, but not substantial on their own.
>
> We agree that L1 loss and bilinear interpolation are not technical novelties by themselves. The paper’s contribution, in this context, is a way to train diffusion models with noisy and incomplete data, where the train-test distribution shift is problematic. We found that a combination of L1, bilinear infilling and step unrolling are effective for diffusion model training with missing data. This enables one to train generic diffusion models that yield remarkably strong performance on both optical flow and monocular depth estimation without the specialized architectures and loss functions that have been common in SOTA models to date.
>
> > Questions:
>
> Q1: Thank you for the suggestion. We will write this more clearly when revising the paper.
>
> Q2: We have included extra examples in Figure 1 in the attachment. We will include more examples in the final version of the paper.

---

> > ### Comment · Reviewer_LqWs · 2023-08-15
> > **Reviewer Response to Rebuttal**
> >
> > Thanks to the authors for the rebuttal.
> >
> > After reading the rebuttal and other reviews I will keep my rating at 7 - accept. I believe this paper should be accepted because it (1) introduces a simple but interesting method that performs SOTA on competitive benchmarks and (2) is very polished and experiments clearly defend all main contributions of the method.

---

### Official Review · Reviewer_F9qi · 2023-07-05

**Soundness:** 3 good
**Presentation:** 3 good
**Contribution:** 4 excellent
**Rating:** 7
**Confidence:** 5

**Summary:**

This paper proposes to use diffusion models to solve monocular depth and optical flow estimation tasks. Unlike previous task-specific models for depth and flow, this paper uses a generic diffusion model. This paper studies the effect of training data (synthetic and real) and processing of sparse depth and flow ground truth when training the diffusions models. Experiments are conducted for depth and flow tasks on representative benchmarks, the proposed method achieves state-of-the-art depth performance on NYU and state-of-the-art optical flow performance on KITTI.

**Strengths:**

- **The idea of using diffusion models to solve depth and optical flow tasks is interesting.** Depth and optical flow are typically approached as regression tasks, it's unclear how the popular diffusion models will perform for both tasks. This paper explores this direction and shows some interesting results.
- **Several strategies are proposed to handle the issue of data for training diffusion models.** It's not straightforward to apply diffusion models to depth and flow tasks, the paper proposes data infilling, step-unrolling and an L1 loss to tackle the challenges.
- **The experiments are extensive and informative.** Training data plays a significant role in training diffusion models, this paper studies the effect of different training datasets for both depth and flow tasks. The performance on KITTI for optical flow task is especially strong, outperforming previous 2-frame optical flow methods by a large margin.
- **A detailed discussion of limitations is presented in the supplementary material.** This paper gives a deep analysis of the limited performance on Sintel test set and the results indicate that a particular sequence on the test set severely affects the averaged performance, which might provide some hints for further improvement in future.

**Weaknesses:**

I didn't observe major weakness and would put some minor points to the Questions.

**Questions:**

- Due to the uncertainty in diffusion models, a same model might produce different results when running twice. How large is the fluctuation and how's the final quantitative results reported? Are the authors using some averaging?
- How sensitive of the proposed architecture to different image resolutions for both depth and flow tasks? For example, what if the inference image resolution is different from training, will the model still perform reasonably?
- The optical flow results on KITTI is very strong, but the results on Sintel seem less robust (as also analysed in the supplementary material). What might cause the different behaviours on Sintel and KITTI, could the authors further comment on this?
- I think one key message from this paper is that the experiments show the importance of training data. When comparing Table 1 and Table 6 for the results of RAFT and the proposed method, we can observe that RAFT outperforms diffusion models when only using AutoFlow for pre-training. However, diffusion models perform better when more datasets are added to the pre-training stage. I am wondering whether this indicates that diffusion models can benefit more from larger datasets than previous regression methods?

**Limitations:**

Yes, the authors have carefully analyzed the limitations.

---

> ### Author Rebuttal · Authors · 2023-08-10
>
> Thank you for your review, and for your thoughtful comments and questions. They will certainly help improve the revised paper. Please see our response below.
>
> > Due to the uncertainty in diffusion models, a same model might produce different results when running twice. How large is the fluctuation and how's the final quantitative results reported? Are the authors using some averaging?
>
> Figure 1, 2, 3, and 4 in the supplemental show multiple samples from the predictive posterior, along with variance heat maps. As visualized in the variance heat maps, the magnitude of local fluctuation depends on the nature of the multi-modality; variability is common in regions of ambiguity (e.g. transparent/reflective surfaces, object boundaries, or occlusions).
> We do average multiple samples for both depth and flow (see Section 4.1). Table 3 shows small but consistent improvements in depth estimation as we average more samples. For optical flow, we average 8 samples for the coarse-grained estimate, but we do not use sample averaging in the high resolution refinement. We will clarify this in the final version of the paper.
>
> > How sensitive of the proposed architecture to different image resolutions for both depth and flow tasks? For example, what if the inference image resolution is different from training, will the model still perform reasonably?
>
> Like prior regression methods (Figure 4 of [17]) our model's performance degrades if one naively runs inference at a resolution that is different from the training resolution. However, in the paper we explain how to use the diffusion model within a coarse-to-fine refinement scheme (see Section 3.3). This way we are able to effectively run inference on high resolution images, first at a coarse-resolution, and then patch-wise at high resolution, conditioned on the coarse-grained estimate to provide global context. Table 5 shows the improved performance on optical flow estimation with this approach.
>
> > The optical flow results on KITTI is very strong, but the results on Sintel seem less robust (as also analysed in the supplementary material). What might cause the different behaviours on Sintel and KITTI, could the authors further comment on this?
>
> In Table 9 of the supplementary material, we show that we outperform FlowFormer on all Sintel test sequences but one, ambush_1. This sequence has significant ambiguity due to an all-white background, large camera and object motions, and the existence of numerous objects that occlude one another and move out of frame. This forces the model to depend heavily on inductive biases and learned motion priors, which is challenging for our method. It is possible that FlowFormer’s inductive bias (with the cost volume etc) make it better suited to reason about this sequence’s large out-of-frame motions.
>
> > RAFT outperforms diffusion models when only using AutoFlow
>
> As discussed in Section 3.1 (paragraph 2) and shown in Figure 3, we find that when trained solely on AutoFlow the diffusion model learns to reproduce shapes from the AutoFlow data, yielding poorer qualitative and quantitative performance. The denoiser’s bias toward
> polygonal shapes in AutoFlow could partially be explained by recent work (eg section 4.2 of [18]) on shape vs texture bias in neural classifiers.  As a result, using AutoFlow alone causes the model to hallucinate on real data and try to identify the best AutoFlow shapes to represent the real objects. Interestingly, this problem is solved by training with larger, more diverse data.
>
> > I am wondering whether this indicates that diffusion models can benefit more from larger datasets than previous regression methods?
>
> This is a great question! The finding that diffusion models benefit more from larger datasets is indeed surprising. Like most regression based flow networks, RAFT has several architectural elements which bias the network towards modeling flow; eg, RAFT features an all-pairs cost volume, which compares all the pixels (in encoding space) in frame 1 to frame 2, and then accesses this cost volume through a lookup operation based on the flow. This network element strongly encourages the network to use pixel comparisons to generate the predicted flow. As a result, despite being trained on only AutoFlow (a synthetic dataset), RAFT can quickly learn to use pixel or patch similarity to compute optical flow, an ability which generalizes quickly to real world videos. In contrast, our diffusion pipeline lacks any of these specific model biases and must learn these biases through data. However, it is important to note that RAFT's ability to learn quickly comes with a tradeoff; namely, the inductive biases that are hard coded into its architecture may be suboptimal. Given enough data, learning these biases may be better than designing them through manual architecture design.

---

> > ### Comment · Reviewer_F9qi · 2023-08-11
> >
> > Thanks for the detailed responses. Overall I agree with the authors, but would like to have a followup discussion on one small point.
> >
> > Regrading the limited performance on the Sintel ambush_1 sequence, the authors mentioned that "FlowFormer’s inductive bias (with the cost volume etc) make it better suited to reason about this sequence’s large out-of-frame motions". However, I guess the cost volume will also be less effective for out-of-frame motions since the points are not matchable across frames? Thus it seems less likely that encoding the matching cost in the architecture will solve this problem? Maybe leveraging some context information (e.g., motion smoothness/propagation in GMA and GMFlow) would be helpful? I think this is an open question and could be considered as future work.
> >
> > This question doesn't affect my opinion, I am happy to accept this paper. Thanks.

---

> > > ### Author Response · Authors · 2023-08-14
> > >
> > > Thank you for your comment. We agree with the reviewer that motion propagation and global motion aggregation are likely the main signals needed. We provide more details below.
> > >
> > > Broadly, we see two categories of errors: (1) From out-of-frame motion. As the reviewer suggests, the motion smoothness/propagation priors used in existing works might help with this. (2) Inconsistent flow for texture-less background regions. For example, ambush_1 frame_1 has a foreground that splits the background into two, with texture on one section of the background and almost no texture on the other. This leads to an ambiguity with our model sometimes predicting consistent flow for the entire background but often not (note that for frames with less ambiguity, the diffusion model is able to successfully propagate the motion). We find that the RAFT model also struggles on such sequences, which suggests that smoothness is not a sufficient prior for handling such cases. As the reviewer mentioned, GMA and GMFlow are some of the first models to successfully address this example (and furthermore the ambush_1 sequence) and it seems likely that this is due to their modules for global motion aggregation and motion propagation, since these signals can align the flow in the background. Theoretically, the self attention blocks in our model should be capable of global motion aggregation so it is possible that our training data distribution does not sufficiently cover such scenes. We hypothesize that there may be multiple ways to solve this problem (data, model design, improvements to sampling, i.e., better approaches to aggregation or re-ranking) and are excited to explore this in future work.
> > >
> > > The reason that the parenthetical in our original response starts with cost volume is because in current models the cost volume is the starting point for global motion aggregation modules (for FlowFormer, the aggregation happens as a post processing of the cost volume into a "cost memory", for GMA, the cost volume is processed into the "motion feature encoder" and then again into the "global motion aggregation" module), but the reviewer is correct to point out that the exact mechanism is global motion aggregation / motion propagation, which should have been mentioned.

---

### Official Review · Reviewer_xrrD · 2023-07-08

**Soundness:** 4 excellent
**Presentation:** 4 excellent
**Contribution:** 3 good
**Rating:** 8
**Confidence:** 3

**Summary:**

The authors study the use of diffusion models for the tasks of single-image depth estimation and optical flow estimation. Self-supervised pre-training, supervised fine-tuning with synthetic and real data, combined with a couple of tricks to leverage imperfect GT, lead to competitive results with nearly no task-specific modifications to the diffusion models. The authors also demonstrated unique capabilities enabled by the diffusion models, e.g., capturing multimodality and completion from partial data.

**Strengths:**

**Originality and significance**:

As far as I know, this is the first work to study the use of generative diffusion models for optical flow and depth estimation tasks. Training good optical flow and depth models typically requires extensive task-specific knowledge in terms of architecture designs and loss functions, so the authors' finding that diffusion models can compete well on these tasks with almost no task-specific treatments is non-trivial and valuable. The competitive results in various settings further add to the significance of the work.

**Technical quality**:

Training diffusion models successfully involves many technical details. The authors generally follow best practices and propose reasonable solutions to unique challenges. More specifically, the use of pre-trained PALETTE, further supervised pre-training with mixtures of synthetic and real datasets, addressing imperfect GT with infilling and step-unrolling, etc. are all well-motivated and proven effective.

**Writing quality**:

The paper is nicely written, with precise language, adequate details, and clear explanations. Conclusions are justified with plenty of results, visualizations, ablation studies, and overall convincing.

**Weaknesses:**

The authors claim that diffusion models can be a generic framework for vision dense prediction tasks, but only consider the tasks of depth estimation and optical flow in this work. Both these tasks are relatively low-level, and it'd be interesting to offer some insights, discussions, or analyses regarding how higher-level tasks, such as semantic segmentation, differ from them and if the claim still holds.

Two relatively minor complaints/suggestions:
* Can authors provide some theoretical analysis or proof for the step-unrolling step? Could unrolling more steps bring further improvements?
* It's unclear how much the use of pallete self-supervised pre-training help since it's not part of the ablation study.

**Questions:**

Please refer to the three points in the weakness section above.

**Limitations:**

I agree with the limitation the authors brought up in the supplementary. Optical flow and depth estimation are low-level tasks commonly used at early stages of real-world application systems and therefore demand higher efficiency. As the authors already pointed out, this is where the proposed diffusion models fall short.

---

> ### Author Rebuttal · Authors · 2023-08-10
>
> Thank you for your review, and for your thoughtful comments and questions. They will certainly help improve the revised paper. Please see our response below.
>
> > Do the claims hold on higher-level tasks such as semantic segmentation
>
> The extent to which a generic diffusion model is effective on other vision tasks, including higher-level tasks is a topic of ongoing work. Diffusion models have indeed been shown to work well on panoptic segmentation [15], which is encouraging.
>
> > Theoretical analysis or proof for step-unrolling
>
> One can view an unrolling step at time t as a Langevin update of a MCMC sampler for which the target distribution is the marginal distribution of the latent $y_t$ (i.e., the distribution of noisy optical flow fields or depth maps). From this perspective unrolling steps act like corrector steps in the predictor-corrector sampler of Song et al [16]. We will include further discussion in the paper.
>
> > Could unrolling more steps bring further improvements
>
> Great question. For datasets in which the ground truth flow or depth maps have more missing data (e.g., like KITTI) one would expect more unrolling steps to be useful in matching the marginal distribution of the latent $y_t$. In our experiments we did find this to be the case (please see Table 2 in attachment).  For instance, increasing the number of unrolling steps from one to four improves KITTI REL from 0.056 to 0.053, and the RMS from 2.700 to 2.568. On NYU, improvements are marginal (REL improves from 0.075 to 0.074 and RMS from 0.324 to 0.315) as one might expect since the ground truth data have fewer missing depth values.
>
> > how much the use of pallete self-supervised pre-training help
>
> Please see Tables 5 and 6 in our supplementary material.  They show that self-supervised pre-training clearly improves results for monocular depth estimation. This isn’t entirely surprising since tasks like inpainting and colorization entail some form of ‘semantic understanding’. We expect similar findings for optical flow estimation, but we did not perform this study on optical flow estimation since pre-training is compute intensive.

---

> > ### Comment · Reviewer_xrrD · 2023-08-20
> >
> > Thank you for the detailed responses. It is a very interesting work with valuable insights and practical values. I don't see any major weaknesses in this submission. I maintain my acceptance recommendation and have no further questions.

---

### Official Review · Reviewer_8jCd · 2023-07-09

**Soundness:** 3 good
**Presentation:** 3 good
**Contribution:** 2 fair
**Rating:** 6
**Confidence:** 3

**Summary:**

The paper demonstrates that diffusion models are effective general-purpose solutions for dense optical flow and monocular depth regression tasks. The paper shows that the same architecture and loss functions lead to at-par or better performance on these tasks, compared to existing methods that use domain knowledge and problem-specific architectures. The main insights presented are the use of a pre-training phase for higher quality, and imputation of the missing values, and a step-unrolled diffusion step, for dealing with incomplete GT training data.

**Strengths:**

The paper is well-written and motivates the contributions perfectly. The simplicity of the architecture and the loss functions are very clear. The results support the claims and show at-par or better results than the state of the art. Extensive experiments on different real datasets help evaluate the method's quality. Multi-modality of the outputs is promising! I would have loved to see some more analysis there.  The presented application is also exciting, showing directions for text to 3D reconstructions.

**Weaknesses:**

While there is limited technical novelty, this is a good paper that demonstrates a simple method for solving two different regression tasks.

- I did not understand why the pre-training phase needs to be separate. The method uses supervised pre-training, that is different for each task. The loss functions are also identical between the pre-training and the fine-tuning states. Why not combine all available datasets and just train the model once (inc. all the tricks used for fine-tuning)? I did not understand this distinction between phases, especially when the experiments are explained and the datasets keep moving from one phase to the other.

- Step-unrolled diffusion is a little similar to self-guidance, introduced in "Analog Bits: Generating Discrete Data using Diffusion Models with Self-Conditioning" [Chen et al.] I wonder if self-guidance is already good at addressing the domain shift problem, or whether step-unrolled diffusion is really needed.

**Questions:**

- Please answer the questions raised on pre-training, and on self-guidance.
- Is it possible to fine-tune an existing diffusion model, such as stable diffusion, instead of training one from scratch? Could it help avoid the use of synthetic data?

**Limitations:**

Yes.

---

> ### Author Rebuttal · Authors · 2023-08-10
>
> Thank you for your review, and for your thoughtful comments and questions. They will certainly help improve the revised paper. Please see our response below.
>
> > limited technical novelty
>
> One of the main motivations for this paper is to understand how well vanilla diffusion models perform on classical dense computer vision tasks which are traditionally solved using specialized techniques. As a result, we intentionally kept the network design and diffusion formulation simple.
>
> We do however, identify and solve several issues unique to the application of diffusion models for the tasks of optical flow and monocular depth estimation. For example, innovations in training are necessary for the application of diffusion models on dense regression problems with noisy and incomplete ground truth data. To that end, we introduce infilling and step-unrolling, which greatly  improves the performance.  For optical flow on the KITTI dataset, baseline diffusion tends to diverge and SOTA performance only results from the combination of both these additions. For KITTI depth, infilling and unrolling result in a substantial improvement in REL (0.222 to 0.056). In addition, we find that our optical flow network requires different training data than regression based optical flow techniques and solve this through a new training regime. Beyond these innovations, we also  introduce a coarse-to-fine refinement scheme which provides further performance gains and greater flexibility in the image resolutions to which the model can be applied. We also demonstrate the use of imputation for text-to-3D generation.
>
> But the main novelty in the paper is the demonstration that it is possible to generate SOTA results on well studied regression problems for which previous methods have relied heavily on specialized techniques such as the use of cost volumes [1] for flow and binning [14] for depth. We propose a common architecture and training procedure across two different dense vision problems. We think this framework is encouraging and motivates new directions for vision research, with a common architecture for many vision problems. Further we show that the diffusion framework is sufficiently powerful to capture the multi-modal predictive distributions (e.g. capturing ambiguity in flow or depth estimation).
>
> > More analysis of multi-modality
>
> The ability of diffusion models to represent complex multimodal distributions, without excessive mode collapse is arguably one of the key properties that has led to the recent success and excitement around diffusion models. Their ability to capture uncertainty in the predictive posterior distributions over depth and optical flow, including multi-modality in cases of ambiguity, is quite interesting. Figures 1-4 in the supplementary material show examples and variance heat maps of the multi-modal predictions for both depth and flow. For depth we observe multimodality in transparent and reflective surfaces, such as mirrors and windows of cars, and around object boundaries. For flow we observe multimodality in transparent surfaces and shadows, thereby capturing the layered nature of the scene. Notably we also observe multimodality in out-of-frame motion where the flow is ambiguous. We think this is an interesting finding and opens up further avenues of research and also provides a way to measure uncertainty which can be important for downstream applications.
>
> > I did not understand why the pre-training phase needs to be separate.
>
> Model pre-training on large-scale datasets is commonplace for optical flow [1, 2] and monocular  depth  [3, 4]. E.g. for optical flow the training data schedule (e.g. FlyingChairs -> FlyingThings3D -> mixture of Sintel / KITTI / VIPER etc.) has been heavily studied and shown to be crucial to achieve good performance (Table 5 of [12] and Table 1 of [13]).  Considerable effort has recently been spent designing better pre-training datasets [10, 11]. This combination of pre-training and fine-tuning provides the advantages of large-scale datasets, with pre-trained models often yielding good zero-shot performance, with the ability to fine-tune models to a specific dataset to maximize performance (perhaps with some loss of generality).
>
> > I wonder if self-guidance is already good at addressing the domain shift problem
>
> In our experiments with self-guidance we found that it does not address the domain shift problem (see Table 1 in attachment). This is expected because, even with self-guidance, there remains a training/inference distribution shift for the latent $y_t$, which is what we address through step-unrolled diffusion training.
>
> > Is it possible to fine-tune an existing diffusion model, such as stable diffusion, instead of training one from scratch? Could it help avoid the use of synthetic data?
>
> Indeed, we use an existing self-supervised pre-trained diffusion model (i.e., the Palette model [5]) for image to image translation. As shown in Tables 5 and 6 of our supplementary material, this self-supervised pre-training substantially improves results.
>
> It may also be possible to use a pre-trained text-conditional image generation model, especially in light of recent works such as [8, 9], which exploit features from pretrained text-to-image models for depth regressors. We briefly considered fine-tuning existing text-to-image models but found that available pixel-space models at the time, such as Imagen [7], were computationally expensive (2B params in the base model) which might limit their utility in practical vision applications and latent space models like Stable Diffusion additionally require dealing with holes in the autoencoder training (like in [6]). Hence, we left this exploration to future work.
>
> It may be possible that the use of significantly more text-image training data or larger datasets for image to image translation would allow one to avoid training with synthetic data. We have not explored this but we agree that this is important future work.

---

> > ### Comment · Reviewer_8jCd · 2023-08-21
> >
> > Thank you for the very well-written rebuttal. It clarifies the contributions and the other questions about the baselines and training details.

---

### Author Rebuttal · Authors · 2023-08-10

References

[1] RAFT: Recurrent All-Pairs Field Transforms for Optical Flow, Teed and Deng, 2020

[2] FlowFormer: A Transformer Architecture for Optical Flow, Huang et al, 2022

[3] Towards Robust Monocular Depth Estimation: Mixing Datasets for Zero-shot Cross-dataset Transfer, Ranftl et al, 2020

[4] Vision Transformers for Dense Prediction, Ranftl et al, 2021

[5] Palette: Image-to-Image Diffusion Models, Saharia et al, 2022

[6] All in Tokens: Unifying Output Space of Visual Tasks via Soft Token, Ning et al, 2023

[7] Photorealistic Text-to-Image Diffusion Models with Deep Language Understanding, Saharia et al, 2023

[8] Unleashing Text-to-Image Diffusion Models for Visual Perception, Zhao et al, 2023

[9] Beyond Surface Statistics: Scene Representations in a Latent Diffusion Model, Chen at al, 2023

[10] AutoFlow: Learning a better training set for optical flow, Deqing et al 2021

[11] Self-supervised AutoFlow, Huang et al, 2023

[12] Models Matter, So Does Training: An Empirical Study of CNNs for Optical Flow Estimation, Sun et al, 2018

[13] FlowNet 2.0: Evolution of Optical Flow Estimation with Deep Networks, Ilg et al, 2017

[14] AdaBins: Depth Estimation using Adaptive Bins, Bhat et al, 2020

[15] A Generalist Framework for Panoptic Segmentation of Images and Videos, Chen et al, 2023

[16] Score-based Generative Modeling through Stochastic Differential Equations. Song et al, 2021

[17] Vision Transformers for Dense Prediction, Ranftl et al, 2021

[18] Text-to-Image Diffusion Models are Zero-Shot Classifiers, Clark and Jaini, 2023

[19] Diffusion Models Beat GANs on Image Synthesis, Dhariwal et al, 2021

---

### Decision · Program_Chairs · 2023-09-21

**Decision:**

Accept (oral)

**Comment:**

The paper has achieved a consensus for acceptance after the rebuttal. The technical contribution, which demonstrates that image diffusion models excel in handling general-purpose dense prediction tasks, is clearly articulated. All reviewers are positive about both the technical novelty and the results presented. The rebuttal addressed most minor concerns and clarity issues; therefore, acceptance should be granted without further discussion. Particularly due to its novelty and the rich insights provided, the AC recommends this paper for a spotlight.